# A δ-cell subpopulation with a pro-β-cell identity contributes to efficient age-independent recovery in a zebrafish model of diabetes

Claudio Andrés Carril Pardo[1†], Laura Massoz[1†], Marie A Dupont[1†], David Bergemann[1], Jordane Bourdouxhe[1], Arnaud Lavergne[1,2], Estefania Tarifeño-Saldivia[1,3], Christian SM Helker[4], Didier YR Stainier[4], Bernard Peers[1], Marianne M Voz[1], Isabelle Manfroid[1]*

[1]Zebrafish Development and Disease Models laboratory, GIGA-Stem Cells, University of Liège, Liège, Belgium; [2]GIGA-Genomics core facility, University of Liège, Liège, Belgium; [3]Gene Expression and Regulation Laboratory, Department of Biochemistry and Molecular Biology, University of Concepción, Concepción, Chile; [4]Department of Developmental Genetics, Max Planck Institute for Heart and Lung Research, Bad Nauheim, Germany

**Abstract** Restoring damaged β-cells in diabetic patients by harnessing the plasticity of other pancreatic cells raises the questions of the efficiency of the process and of the functionality of the new *Insulin*-expressing cells. To overcome the weak regenerative capacity of mammals, we used regeneration-prone zebrafish to study β-cells arising following destruction. We show that most new *insulin* cells differ from the original β-cells as they coexpress Somatostatin and Insulin. These bihormonal cells are abundant, functional and able to normalize glycemia. Their formation in response to β-cell destruction is fast, efficient, and age-independent. Bihormonal cells are transcriptionally close to a subset of δ-cells that we identified in control islets and that are characterized by the expression of *somatostatin 1.1* (*sst1.1*) and by genes essential for glucose-induced Insulin secretion in β-cells such as *pdx1*, *slc2a2* and *gck*. We observed in vivo the conversion of monohormonal *sst1.1*-expressing cells to *sst1.1+ ins +* bihormonal cells following β-cell destruction. Our findings support the conclusion that *sst1.1* δ-cells possess a pro-β identity enabling them to contribute to the neogenesis of Insulin-producing cells during regeneration. This work unveils that abundant and functional bihormonal cells benefit to diabetes recovery in zebrafish.

**\*For correspondence:**
Isabelle.Manfroid@uliege.be

[†]These authors contributed equally to this work

## Editor's evaluation

Recently, there has been a growing appreciation for the existence of cellular plasticity in the adult islet. This study probes this phenomenon by exploiting the zebrafish experimental model, which has a much higher adult regeneration capacity than in mammals. A particular novel finding from the study is the identification of a subpopulation of islet delta cells that are similar to beta cells at the transcriptional level and can convert into an insulin/somatostatin co-expressing cell population upon beta cell ablation. The findings will be of particular interest to researchers interested in islet cell biology and pancreatic endocrine cell reprogramming; it will be interesting to explore whether similar delta cell subpopulations exist in mammalian islets to serve as an alternative source of insulin producing cells.

## Introduction

Insulin-producing β-cells reside in pancreatic islets where they are intermingled with other endocrine cells such as α-cells, secreting glucagon (Gcg), and δ-cells secreting somatostatin (Sst). Elevation of extracellular glucose concentration triggers glucose uptake by β-cells through the glucose transporter GLUT2 (*Slc2a2*). Glucose is then metabolized to generate ATP which will trigger the closure of the $K_{ATP}$ channel formed by Kir6.2 (*Kcnj11*) and SUR1 (*Abcc8*), membrane depolarization, $Ca^{2+}$ influx and release through exocytosis of insulin secretory granules into the blood. In mature β-cells, this process is further amplified by other molecules such as amino acids, fatty acids, hormones (incretins GLP-1, GIP), and neural factors (dopamine, adrenaline…) via the cAMP messenger. Dysfunction of these processes leads to impaired insulin secretion, chronic hyperglycemia, and diabetes. In Type 2 diabetes, chronic glucolipotoxic stress ultimately provokes β-cell failure and death. In Type 1 diabetes, on the other hand, the destruction of β-cells is mediated by autoimmune attack.

Human adult β-cells are quiescent and barely possess the capacity to compensate their destruction through increased proliferation. Alternative mechanisms inferred from studies in mice revealed the striking plasticity of other pancreatic endocrine cell types towards the β-cell phenotype. For example, Ins+ Gcg + bihormonal cells form after acute β-cell destruction mediated by transgenic expression of the diphteria toxin receptor (DTR) in adult mice (*Thorel et al., 2010*). These cells derive from a small fraction of α-cells that switch on the β-cell markers Pdx1, Nkx6.1 and Ins through direct conversion, leading to restoration of about 10% of the β-cell mass after 10 months. As this process is quite slow and inefficient, adult DTR mice do not survive without injection of insulin during the first months after ablation. In contrast, at juvenile stages, β-cell neogenesis occurs from transdifferentiation of δ-cells (*Chera et al., 2014*). In this case, δ-cells dedifferentiate, lose *Sst* expression, replicate and redifferentiate into β-cells. About 23% of the initial β-cell mass has recovered 4 months after ablation emphasising faster and more efficient improvement of glycemia than in adult mice. Very recently, a rare population of pancreatic polypeptide (Ppy)-expressing γ-cells has also been shown to display plasticity and to activate *Ins* expression in response to β-cell injury (*Perez-Frances et al., 2021*). Hence, various pancreatic islet cells possess a remarkable plasticity yet the regeneration potential is generally limited in adult mammals.

In contrast to the limited regeneration capacity of adult mammals, zebrafish are notorious for their potent, spontaneous and rapid regeneration of β-cells from larval to adult stages (*Curado et al., 2007*; *Delaspre et al., 2015*; *Ghaye et al., 2015*; *Moss et al., 2009*; *Ninov et al., 2013*; *Pisharath and Parsons, 2009*; *Ye et al., 2015*). In zebrafish, α-cells transdifferentiate into Ins-expressing cells after β-cell destruction (*Ye et al., 2015*). On the other hand, unlike mouse models in which regeneration via progenitors or precursors is debated, β-cell neogenesis is well recognised in zebrafish to involve regenerative processes from progenitor-like cells present in the ducts (*Delaspre et al., 2015*; *Ghaye et al., 2015*; *Ninov et al., 2013*). β-cell destruction is accomplished in zebrafish using a chemo-genetic system based on the transgenic expression of the bacterial nitroreductase (NTR) under the control of the *ins* promoter where cell death is induced by a nitroaromatic prodrug (*Bergemann et al., 2018*; *Curado et al., 2007*; *Pisharath and Parsons, 2009*). In adults, after a huge rise of glycemia within 3 days, the pancreas is replenished with new β-cells in 2–3 weeks which correlates to a return to normoglycemia.

De novo formation of β-cells in order to repair damaged islets constitutes a promising therapeutic perspective for diabetic patients. However, new β-cells could show differences in their number and identity impacting on their activity. For example, the presence in mice of Gcg+ Ins + cells, although apparently functional, should be considered cautiously as inappropriate differentiation of β-cells and impaired maturation or identity are common shortcoming in diabetes (*Moin and Butler, 2019* for review).

Using the larval and adult zebrafish as regeneration models, we investigated the identity of regenerated β-cells and discovered that most new *ins*-expressing cells are Ins + Sst1.1+ bihormonal cells. We identified a specific δ-cell subpopulation distinct from the previously identified zebrafish *sst2* δ-cells that is characterized by the expression of *sst1.1* and of several important β-cell features. The transcriptomic profile of bihormonal cells is also very close to the *sst1.1* δ-cells, making them resemble β/δ hybrids. By in vivo imaging of larvae, we observed the appearance of *ins*-expressing bihormonal cells from monohormonal *sst1.1* δ-cells early after β-cell ablation. We also provide evidence that pancreatic ducts contribute to the pool of bihormonal cells. Furthermore, bihormonal cells are abundant in the

regenerated pancreas and able to normalize glycemia after a glucose challenge. Our findings show the importance of bihormonal cells in the spontaneous recovery of diabetic zebrafish.

## Results

### Most regenerated β-cells in adult zebrafish coexpress Ins and Sst

To characterize the new β-cells after regeneration, we used 6- to 10-month-old *Tg(ins:NTR-P2A-mCherry)* (*Bergemann et al., 2018*) adult fish to first ablate β-cells. Basal blood glucose was monitored to evaluate ablation (3 days post treatment, dpt) and regeneration (20 dpt). As expected, fasting basal blood glucose dramatically raised at three dpt compared to CTL fish which reflected efficient ablation (*Figure 1A* and *Figure 1—source data 1*). After 20 days, glycemia was impressively improved though still slightly above control values. A preliminary RNAseq experiment on mCherry+ cells isolated from the main islet of *Tg(ins:NTR-P2A-mCherry)* adult fish 2 months after ablation revealed strong expression of the *sst1.1* gene in regenerated β-cells just below *ins* (*Figure 1—source data 3*), thereby suggesting that regenerated β-cells are bihormonal. As blood glucose is nearly normalized after 20 days, we characterized these cells at this time point. Immunofluorescence on regenerated 20 dpt islets showed many Ins + cells that also displayed Sst immunolabeling (*Figure 1B*). In contrast, control islets showed robust staining of the endogenous Ins and Sst hormones without appreciable overlap, thus demarcating monohormonal β- and δ-cells (*Figure 1B*). We next created a *Tg(sst1.1:eGFP)* reporter line driving GFP in *sst1.1*-expressing cells. This transgene was not active in β-cells of control islets (*Figure 1—figure supplement 1*). Similar to what was observed with the endogenous Sst and Ins proteins, regenerated 20 dpt islets of *Tg(sst1.1:eGFP); Tg(ins:NTR-P2A-mCherry)* fish contained many cells coexpressing GFP with mCherry, while GFP and mCherry-labeled distinct cells in control islets (*Figure 1C*). Strikingly, double positive cells could already be detected 3 days after ablation, although they displayed low levels of mCherry.

We next quantified *ins*+ β-cells, *sst1.1*+ cells and double *ins + sst1.1*+ cells by measuring the number of mCherry+, GFP+, and GFP+ mCherry + cells, respectively, in *Tg(sst1.1:eGFP); Tg(ins:NTR-P2A-mCherry)* adult fish. The main islet was obtained by dissection and the different cell populations were analyzed by FACS (*Figure 1D–G*, *Figure 1—figure supplement 2* and *Figure 1—source data 2*). At 3 and 20 dpt, we observed a drastic loss of mCherry+ (GFP-) β-cells with a drop to 3.2% of the initial β-cell mass at 3 dpt (*Figure 1E*). In contrast, a large population of double GFP+ mCherry + cells appeared that represented 43% of the initial β-cell mass (*Figure 1F*). These cells still persisted at 20 dpt and they made up at this stage 98% of the *ins*-expressing cells. At 20 dpt, mCherry+ GFP- β-cells still constituted a very minor population. (*Figure 1E*). After ablation, the amount of GFP+ mCherry cells also decreased (*Figure 1G*).

In conclusion, these results indicate that *ins + sst1.1*+ bihormonal cells rapidly appear in the main islet after β-cell ablation in adult fish and persist steadily for at least 20 days. They constitute the vast majority of the new *ins*-expressing cells following ablation.

### Genesis of bihormonal cells also occurs during regeneration in larval stages and is independent of the ablation model

As in mouse the process of bihormonal cells (in that case Gcg+ Ins + ) formation after β-cell ablation is specific to adult stages (*Thorel et al., 2010*; *Chera et al., 2014*), we next asked whether Sst1.1+ Ins + bihormonal cells also appear in zebrafish larvae. We therefore performed the ablation in *Tg(sst1.1:eGFP); Tg(ins:NTR-P2A-mCherry)* at 3 days post fertilization (dpf) and assessed the expression of *ins*:mCherry and *sst1.1*:GFP. Like in adults, bihormonal cells were detected 3 days after ablation (3 dpt, 6 dpf) (*Figure 2A–B*). We confirmed by in situ hybridization detecting the endogenous mRNAs that these bihormonal cells express *sst1.1* together with *ins* (*Figure 2C*). This experiment also revealed that they do not coexpress *sst2* (*Figure 2C*).

Then we questioned if the bihormonal cells can also be induced using another system of β-cell destruction. We chose the Diphteria Toxin chain alpha (DTA) suicide transgene which has previously been used to efficiently ablate β-cells (*Ninov et al., 2013*). Ablation was achieved in *Tg(ins:lox-mCherry-lox-DTA); Tg(ins:CRE-ERT2)* larvae by performing a 4-OHT treatment at 7 dpf and the larvae

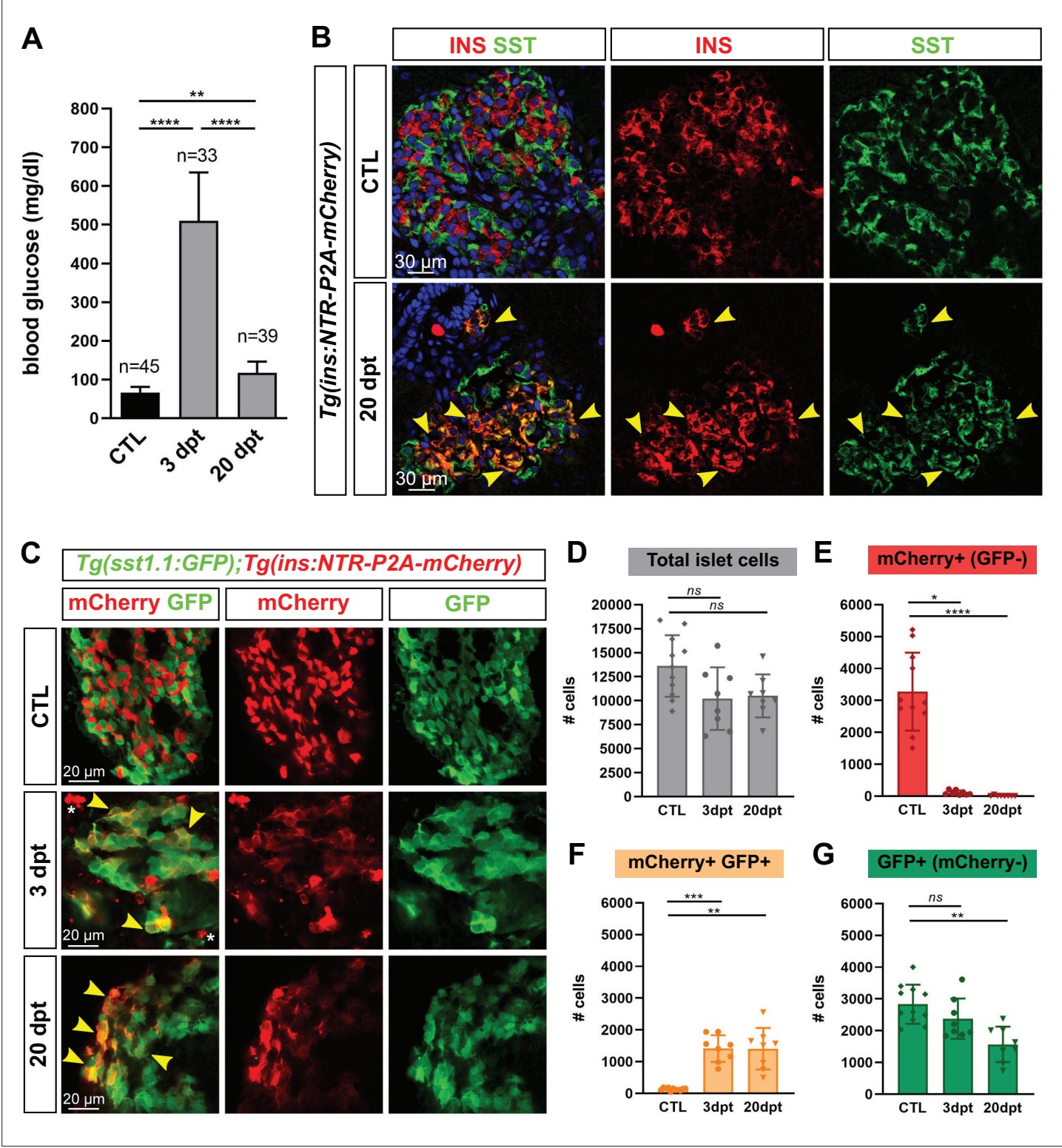

**Figure 1.** Most new ins + cells after ablation and regeneration in zebrafish are Ins + Sst1.1+ bihormonal cells. (**A**) Blood glucose level (mg/ml) of adult *Tg(ins:NTR-P2A-mCherry)* control fish (CTL, 66 ± 15 mg/dl), 3 days (510 ± 126 mg/dl) and 20 days post treatment (dpt) (117 ± 29 mg/dl) with the NFP prodrug to trigger β-cell ablation. The huge rise of glycemia at three dpt confirms the efficiency of ablation. One-way ANOVA Kruskal-Wallis test (with Dunn's multiple comparisons); Mean ± SD; **p < 0.005, ****p < 0.0001. (See *Figure 1—source data 1*). (**B**) Immunolabeling of β- and δ-cells with anti-INS (red) and anti-SST (green), respectively, on paraffin sections through the main islet of *Tg(ins:NTR-P2A-mCherry)* adult fish in control condition (CTL) and at 20 dpt. In CTL islet, no appreciable overlap between the two markers can be detected while broad colabeling is observed

*Figure 1 continued on next page*

*Figure 1 continued*

at 20 dpt and represented by many yellow cells (arrowheads). (**C**) Whole mount immunodetection of β- and *sst1.1+* cells in the main islet of adult *Tg(sst1.1:GFP);Tg(ins:NTR-P2A-mCherry)* fish by labeling with anti-GFP marking *sst1.1*-expressing cells and anti-mCherry for β-cells. Both cell types show no or very few overlapping in CTL fish. At 3 and 20 dpt, many double GFP+ mCherry + cells are observed (yellow cells, arrowheads). Bright mCherry+ β-cell debris are detectable at 3 dpt (white asterisk). (**D–G**) Quantification of the GFP+, mCherry+ (β-cells) and double GFP+ mCherry + cells detected by FACS in the main islets of *Tg(sst1.1:GFP);Tg(ins:NTR-P2A-mCherry)* CTL fish and following β-cell ablation (3 and 20 dpt), based on fluorescence analysis shown in *Figure 1—figure supplement 2*. (**D**) Total islet cell number in CTL, 3 dpt, and 20 dpt islets. (**E**) CTL islets contain 3277 ± 1220 mCherry+ (GFP-) β-cells. At 3 dpt, ablated β-cells represent 105 ± 70 cells and were even more scarce at 20 dpt (14 cells). (**F**) Double GFP+ mCherry + bihormonal cells represent 135 ± 45 cells in CTL islets, 1411 ± 421 cells at 3 dpt and 1409 ± 655 cells at 20 dpt. (**G**) GFP+ (mCherry-) cells represent 2833 ± 615 cells in CTL islets. One-way ANOVA Kruskal-Wallis test (with Dunn's multiple comparison); *ns*, not significant, *p < 0.05, **p < 0.005, ***p < 0.0005, ****p < 0.0001; Mean ± SD (See *Figure 1—source data 2*).

The online version of this article includes the following source data and figure supplement(s) for figure 1:

**Source data 1.** Blood glucose values.

**Source data 2.** Cell quantification values.

**Source data 3.** Top 20 genes expressed in regenerated β-cells.

**Figure supplement 1.** *Tg(sst1.1:GFP)* is active in sst1.1+ cells and not in β-cells.

**Figure supplement 2.** Analysis of sst1.1:GFP and *ins:NTR-P2A-mCherry* fluorescent cells by flow cytometry.

were then analyzed at 16 dpf (*Figure 2—figure supplement 1*). Similar to our observations with the NTR system, Ins and Sst immunofluorescence revealed many coexpressing cells.

In conclusion, these data demonstrate that there is no specific competent stage for the formation of Ins + Sst1.1+ bihormonal cells in zebrafish. In addition, this process does not depend on the method of ablation.

## Most bihormonal cells do not derive from pre-existing β-cells

To explore the possibility that bihormonal cells derive from pre-existing β-cells spared by the ablation, β-cells were traced before ablation using *Tg(ins:CRE-ERT2); Tg(ubb:loxP-CFP-loxP-zsYellow); Tg(sst1.1:GFP); Tg(ins:NTR-P2A-mCherry)* fish. As bihormonal cells were also observed at six dpf, we used larvae to tackle their origin by CRE-mediated recombination (*Hans et al., 2009*; *Mosimann et al., 2011*). We treated the larvae with 4-OHT at six dpf to label the β-cells and performed the ablation the next day (*Figure 2D*). We found that, 7 days after ablation, only 10% of the bihormonal cells were positive for the zsYellow lineage tracer (*Figure 2E–E' , and H*). To ensure that this low level was not due to an inefficient tracing, we checked non-ablated larvae and found that 94% of the β-cells were labeled with zsYellow (*Figure 2F–G*). In addition, the *sst1.1*:GFP+ cells were not labeled (*Figure 2F*). These data demonstrate good efficiency and specificity of the tracing. Based on these observations, we can conclude that some bihormonal cells originate from pre-existing β-cells but the majority arises from non-β origin(s).

## *ins+ sst1.1+* bihormonal cells share similarities with β- and δ-cells, and possess the basic machinery for glucose responsiveness

In order to characterize the *ins + sst1.1+* bihormonal cells after regeneration, we analyzed their transcriptomic profile. To this end, double GFP+ mCherry + cells were isolated by FACS from the main islet of *Tg(sst1.1:eGFP); Tg(ins:NTR-P2A-mCherry)* adult fish at 20 dpt. Control β-cells (mCherry+ GFP-) were obtained from age-matched, non-ablated, transgenic fish. We compared their RNAseq profiles and identified 887 DE genes with a higher expression in bihormonal cells and 705 DE genes higher in β-cells (Padj <0.05 and above twofold differential expression) (*Figure 3A–B* and *Figure 3—source data 1*). In accordance with the weak mCherry fluorescence harbored by GFP+ mCherry + cells as compared to native β-cells, the expression of *ins* in bihormonal cells was fivefold below its typical level in β-cells (*Figure 3C*). Also, as expected, the δ-cell hormone *sst1.1* was sharply overexpressed in bihormonal cells (209-fold) compared to its basal level in β-cells, and was even the top hormone just above *ins* (*Figure 3C*). The other pancreatic hormones known in zebrafish, *sst1.2*, *sst2*, *gcga*, *gcgb*, and *ghrl*, were detected at much weaker levels in both ins + populations (*Figure 3C*). Accordingly, Gcg protein was undetectable in bihormonal cells by immunofluorescence (*Figure 3—figure*

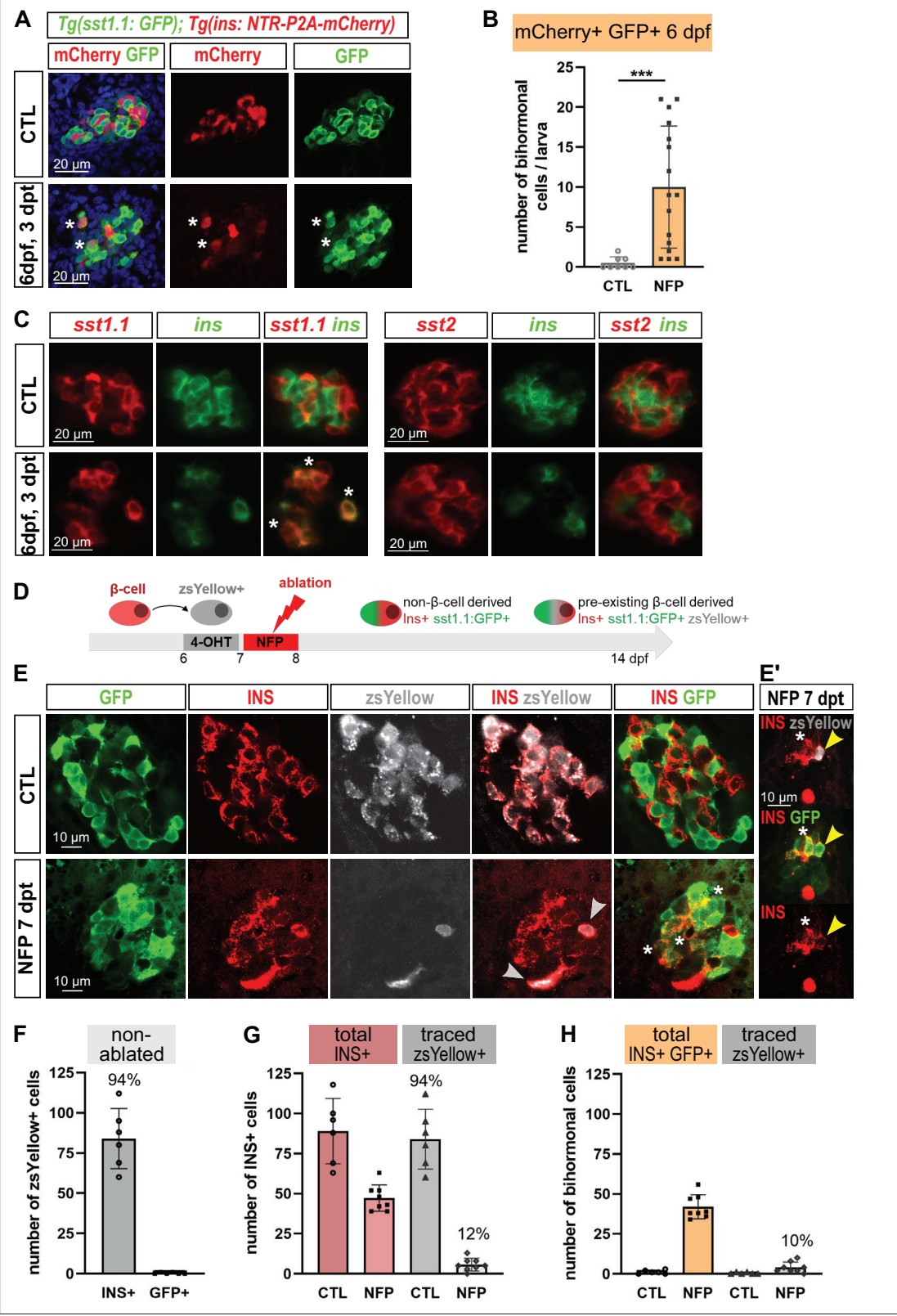

**Figure 2.** Bihormonal cell formation is age- and ablation model-independent and mostly do not derive from escaping β-cells. (**A**) Whole mount immunodetection in six dpf *Tg(sst1.1:GFP); Tg(ins:NTR-P2A-mCherry)* larvae showing β-cells (mCherry, red), *sst1.1*-expressing cells (GFP, green) and double positive bihormonal cells (asterisks) in the main islet in control (CTL) and 3 days after NFP-mediated ablation (3 dpt). Representative confocal images (single optical planes). dpf: days post-fertilization. (**B**) Quantification of bihormonal cells co-labeled by mCherry and GFP based on confocal

*Figure 2 continued on next page*

*Figure 2 continued*

images of 6 dpf larvae. Unpaired two-tailed t-test (with Welch correction); \*\*\*p < 0.001; Mean ± SD. (**C**) Whole mount fluorescent in situ hybridization performed on 6 dpf *Tg(ins:NTR-P2A-mCherry)* larvae with an *ins* antisense RNA probe (green) combined with either a *sst1.1* or a *sst2* probe (red). NFP-mediated ablation was performed from 3 to 4 dpf. Representative confocal images of the main islet (single optical planes). (**D–G**) β-cell tracing with *Tg(ins:CRE-ERT2); Tg(ubb:loxP-CFP-loxP-zsYellow); Tg(sst1.1:GFP); Tg(ins:NTR-P2A-mCherry)* larvae. (**D**) Experimental design: CRE recombination was performed by treatment with 4-OHT treatment at six dpf to induce the expression of the lineage tracer zsYellow (gray) in β-cells (INS, red). β-cell ablation (NFP) was then performed at seven dpf and the lineage tracer was analysed in the main islet at 14 dpf (7 dpt). (**E-E'**) Confocal images showing immunodetection of GFP (green), zsYellow (gray), and INS (red) antibodies. After ablation, traced β-cells are evidenced by double zsYellow+ Ins + staining (gray arrowheads) and bihormonal cells by double Ins + GFP + staining (white asterisks). (**E'**) Close-up showing two bihormonal cells, one zsYellow+ (derived from a pre-existing β-cell) (yellow arrowhead) and one zsYellow- (asterisk). (**F–H**) Quantification (CTL, n = 6; NFP, n = 8) based on the confocal images. (**F**) In CTL non-ablated islets, ZsYellow marked efficiently the Ins+ β-cells (84 ± 19 zsYellow+ Ins + cells out of 89 ± 20 total Ins+ β-cells, representing 94% of the total β-cells). ZsYellow was not detected in sst1.1:GFP+ cells, showing a good specificity. (**G**) 7 days after ablation (NFP), 47.3 ± 8 Ins + cells were detected and 5.8 ± 4 of them (12%) expressed zsYellow. (**H**) 42 ± 7.5 Ins + cells are also GFP+ bihormonal and 10% of them (4 ± 3 cells) are labeled with zsYellow. Mean ± SD.

The online version of this article includes the following figure supplement(s) for figure 2:

**Figure supplement 1.** Bihormonal cell formation following β-cell ablation with Diphteria Toxin A.

supplement 1). Collectively, these data confirm that bihormonal cells coexpress high levels of two main hormones, *ins* and *sst1.1,* at both the mRNA and protein levels.

To further characterize these bihormonal cells, we assessed the expression of transcription factors important for β-cell development and identity in zebrafish and mouse/human (see list in ***Figure 3— source data 5***). We first checked the expression of the pan-endocrine genes *neurod1*, *pax6b* and *isl1* and found similar expression (***Figure 3C***). We also examined the expression of *pdx1*, a transcription factor essential for *ins* expression in β-cells. *pdx1* was equally expressed in both native β-cells and post-regeneration GFP+ mCherry + cells. We next evaluated the β-cell identity of bihormonal cells by interrogating the expression of zebrafish β-cell markers. We defined these markers as genes enriched in β-cells ( > 4 fold) versus the other main pancreatic cell types (α-, *sst2* δ-cells, acinar and ductal cells) based on previous RNAseq data (***Tarifeño-Saldivia et al., 2017***; ***Figure 3—source data 2***). This list of β-cell genes includes *nkx6.2*, a previously identified β-cell marker in zebrafish (***Binot et al., 2010***; ***Tarifeño-Saldivia et al., 2017***) which is the equivalent of *Nkx6.1* in mouse/human β-cells (***Figure 3— source data 5***). More than half of the 62 'β-cell genes' were expressed at similar levels in both bona fide β-cells and post-regeneration bihormonal cells. In contrast, 27 β-cell genes showed either over- or underexpression (***Figure 3D***). In particular, 18 β-cell genes were underexpressed in bihormonal cells like, for example, *nkx6.2* which was not expressed at all (***Figure 3E***). We also looked at markers of dedifferentiation and found that the zebrafish pancreatic progenitor markers *nkx6.1*, *sox9b,* and *ascl1b*, were barely expressed in bihormonal cells, like in control β-cells.

When considering key genes for β-cell function and maturation, that is glucose sensing, uptake, Ins maturation and secretion, many were expressed at comparable levels in both cell types, such as notably *slc2a2*, *pcsk1*, *abcc8*, and *snap25a* (***Figure 3E***). *ucn3l*, a marker of mature β-cells in mammals (***Blum et al., 2012***) and zebrafish (***Singh et al., 2017***), was overexpressed in bihormonal cells.

Gene Ontology (GO) analysis of the genes overexpressed in bihormonal cells compared to β-cells showed that the top significant biological processes were related to adhesion and neuronal synapses with many genes that are known in β-cells to be important for Insulin processing and exocytosis (***Figure 3F–G*** and ***Figure 3—source data 3***). Other processes included intracellular Calcium and cAMP signaling (***Figure 3F–G*** and ***Figure 3—source data 3***). These data strongly suggest that bihormonal cells, like β-cells, are excitable cells with the capacity to secrete Insulin in response to glucose.

Altogether, these data indicate that bihormonal cells possess the molecular bases of functional mature β-cells such as a glucose-responsiveness and hormone secretion machinery. However, although many β-cell genes are similarly expressed between bihormonal and β-cells, bihormonal cells display a divergent identity such as lack of the zebrafish β-cell marker *nkx6.2* and strong expression of *sst1.1*.

## Bihormonal cells constitute the main source of insulin in regenerated zebrafish and restore blood glucose homeostasis

The basal glycemia of regenerated fish is nearly normalized after 20 days, strongly suggesting that bihormonal cells – that represent 98% of the Ins-producing cells – contribute to blood glucose control.

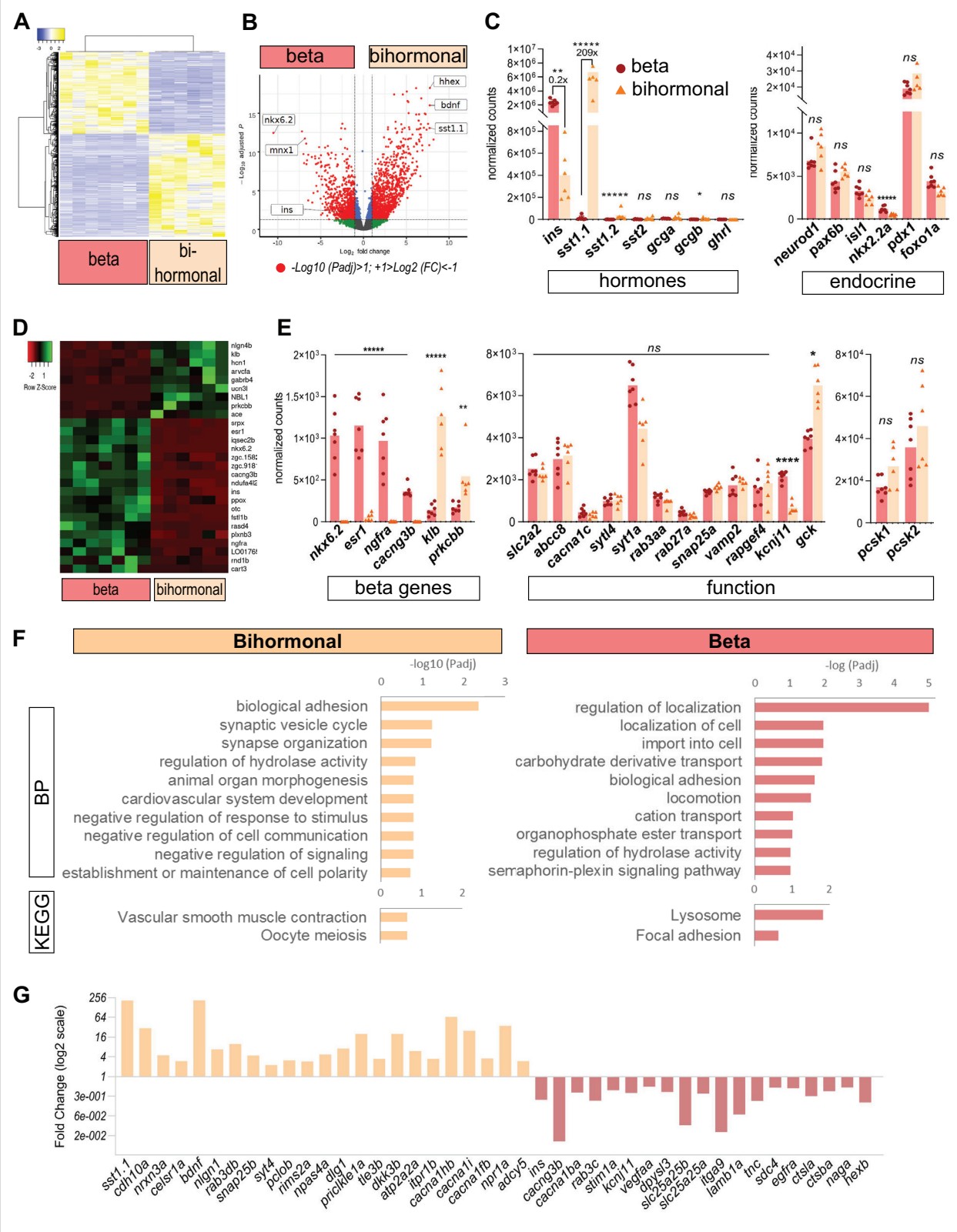

**Figure 3.** Transcriptomic comparison of bihormonal cells and β-cells. (**A**) Heatmap representation of the transcriptomes of 20 dpt bihormonal (six replicates) and β-cells (seven replicates) (significant DE genes). (**B**) Volcano plot showing the distribution of genes in β-cells without ablation and bihormonal cells. The x-axis represents the log$_2$ of fold change (FC) and the y-axis the log$_{10}$ of adjusted P value (Padj) provided by DESeq. The red dots highlight the significantly DE genes (Padj <0.05). A full list of significant DE genes is provided in *Figure 3—source data 1*. (**C**) Expression values (mean

*Figure 3 continued on next page*

*Figure 3 continued*

normalized reads) as provided by DESeq of the main hormones and endocrine genes in β-cell and bihormonal cell transcriptomes. *sst1.1* and *ins* are the two highest expressed hormones. Padj are calculated by DESeq. *ns: no significant DE between the two conditions, 0.05< P* < 0.005, 0.005< P** < 0.0005, P***** < 0.000005.* (**D**) Heatmap plot showing the direction and amplitude of changes in expression of the β-cell markers between normal β-cells and bihormonal cells (significant DEG only). The 62 β-cell markers are provided in *Figure 3—source data 2*. (**E**) Expression values (mean normalized reads) as provided by DESeq of selected β-cell markers and genes important for β-cell function in β-cells and bihormonal cells. Padj are calculated by DESeq. *ns:* no significant DE between the two conditions, 0.05<* < 0.005, 0.005<** < 0.0005, 0.00005<**** < 0.000005, ***** < 0.000005. (**F**) Enriched Gene Ontology (GO) terms. Top 10 or Padj (FDR) < 0.25 Biological Processes (BP) and KEGG pathways are shown. The plots represent the enrichment ratio of Biological Processes and KEGG pathways identified with WebGestalt (*Liao et al., 2019*) using the genes over- and underexpressed in bihormonal cells compared to β-cells obtained with a twofold differential expression and Padj <0.05. All overrepresented Biological Processes and Pathways (< FDR 0.25) are listed in *Figure 3—source data 3* (bihormonal cells) and *Figure 3—source data 4* (β-cells). (**G**) Over- and underexpression of selected significantly DE genes from the BP and KEGG pathways identified in β-cells and bihormonal cells (Fold Change, log2 scale).

The online version of this article includes the following source data and figure supplement(s) for figure 3:

**Source data 1.** Differentially expressed genes between beta cells and bihormonal cells.

**Source data 2.** List of the beta cell markers from *Tarifeño-Saldivia et al., 2017*.

**Source data 3.** Gene Ontology analysis of genes overexpressed in bihormonal cells.

**Source data 4.** Gene Ontology analysis of genes overexpressed in beta cells.

**Source data 5.** Table of the main transcription factors considered in this study, their expression and comparison between zebrafish and mouse/human.

**Figure supplement 1.** Gcg is not detected in bihormonal cells.

To exclude the possibility that glycemia is regulated by a population of genuine monohormonal β-cells regenerated outside the main islet, we analyzed the pancreatic tail. Indeed, zebrafish possess smaller secondary islets scattered in the pancreatic tail in addition to the large main islet located in the head. Similar to the main islets, regenerated 20 dpt secondary islets harbored many bihormonal cells and very scarce monohormonal β-cells (*Figure 4A–B* and *Figure 4—source data 1*). Thus, bihormonal cells constitute the predominant source of Ins throughout the whole pancreas.

To assess the functionality of adult bihormonal cells, we performed a glucose tolerance test and blood glucose levels were followed after an intraperitoneal injection of D-Glucose. Regenerated fish 20 days after β-cell ablation displayed completely normal glucose tolerance (*Figure 4C* and *Figure 4—source data 2*). Together, all these data support the conclusion that the bihormonal cells are responsible for the normalization of glycemia and glucose tolerance in regenerated zebrafish.

## *sst1.1* δ-cells are distinct from *sst2* δ-cells and display similarities with β-cells

Given the expression of *sst1.1* in bihormonal cells, we sought to characterize the *sst1.1*-expressing cells in normal islets without ablation. Previous transcriptomic studies of pancreatic cells detected three *Sst* genes in normal adult pancreatic islets, *sst1.1*, *sst1.2*, and *sst2* (*Spanjaard et al., 2018*; *Tarifeño-Saldivia et al., 2017*). However, so far, only the *sst2* δ-cells, which also express *sst1.2*, have been fully characterized (*Tarifeño-Saldivia et al., 2017*). We thus isolated the *sst1.1*-expressing GFP+ cells from control non-ablated islets of *Tg(sst1.1:eGFP); Tg(ins:NTR-P2A-mCherry)* adult fish to determine their transcriptome. Close examination of these *sst1.1*:GFP+ cells by flow cytometry actually distinguished two subpopulations recognised by different levels of GFP fluorescence, GFP[low] and GFP[high] (*Figure 5—figure supplement 1A*). The GFP[high] population represented 35% of all GFP cells. The presence of cells with high and low GFP were also observed in situ by immunofluorescence on fixed whole pancreas (*Figure 5A*).

The transcriptomic profile of the two GFP populations was obtained (*Figure 5—figure supplement 1B*). Principal Component Analysis (PCA) unveiled that GFP[high] cells are very similar to bihormonal cells (*Figure 5B*). In addition, they are also more similar to β-cells than GFP[low] cells. Clustering analysis of the two GFP populations, the bihormonal cells and the other endocrine cells (a, b, and *sst2* δ-cells *Tarifeño-Saldivia et al., 2017*) also showed that the GFP[high] cells cluster together with bihormonal cells and apart from the GFP[low] cells (*Figure 5C*). Indeed, GFP[low] cells were closer to *sst2* δ-cells than to the other endocrine subtypes. Comparison of the two GFP populations identified 975 and 1206 DE genes overexpressed in GFP[high] and GFP[low], respectively (FC >2, Padj <0.05) (*Figure 5D* and *Figure 5—source data 1*). *sst1.1* was by far the predominant *Sst* gene expressed in GFP[high] cells

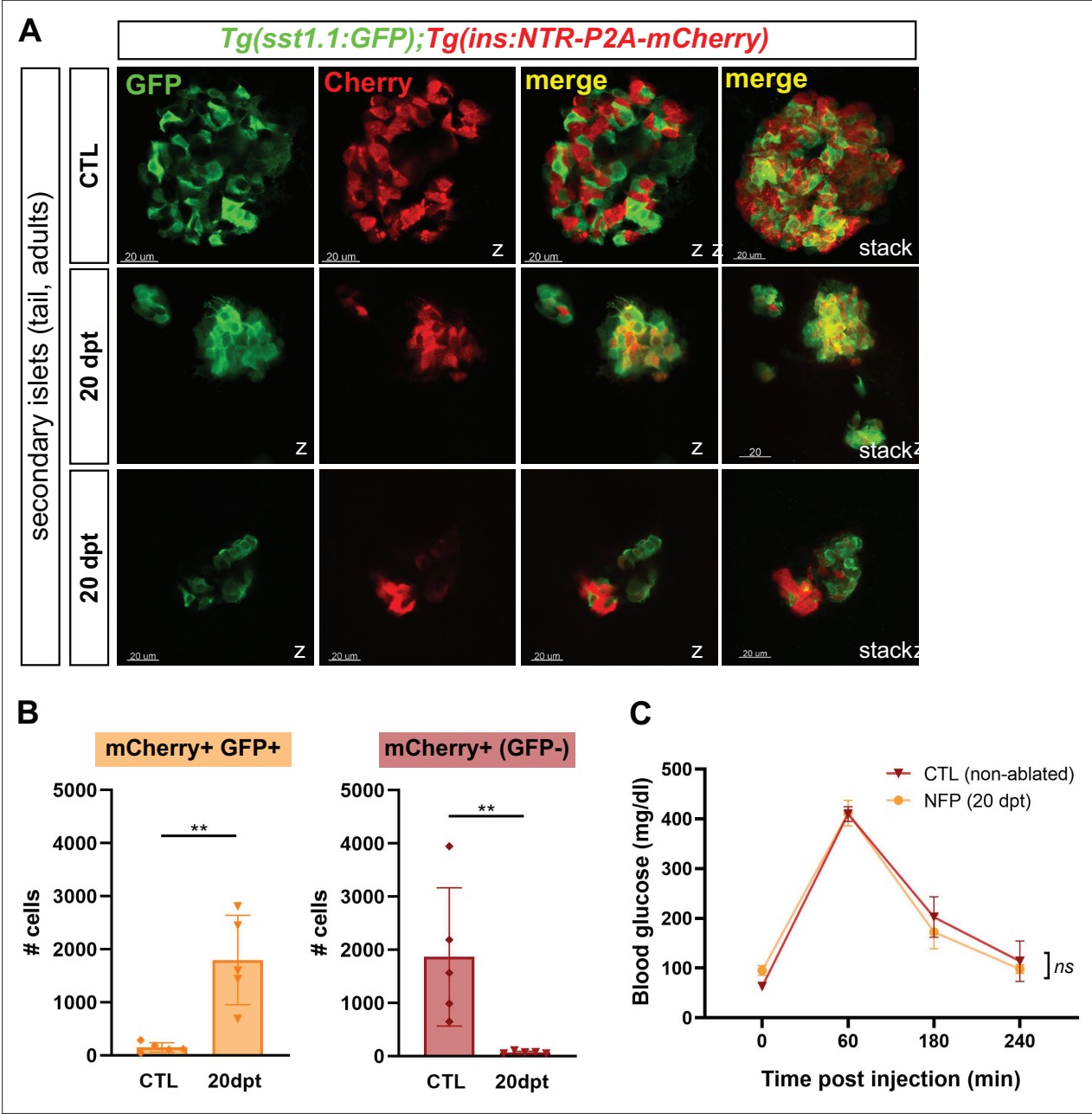

**Figure 4.** Bihormonal cells are the main source of Insulin in the whole pancreas after regeneration and regulate blood glucose homeostasis. (**A**) Whole mount immunofluorescence (GFP and mCherry) on the pancreas of *Tg(sst1.1:eGFP); Tg(ins:NTR-P2A-mCherry)* adult zebrafish showing secondary islets in the pancreatic tail. One representative CTL and two independent 20 dpt samples are shown. Coexpressing cells appear in yellow due to overlapping GFP and mCherry staining. Confocal optical section (Z-planes) and 3D projections (stacks) are shown. (**B**) Quantification of monohormonal mCherry+ β-cells and GFP+ mCherry + bihormonal cells detected by FACS in the tail of CTL fish and after 20 days regeneration (20 dpt). Mann-Whitney test. p** = *0.0079* in both graphs. Mean ± SD. (See also *Figure 4—source data 1*). (**C**) Intraperitoneal glucose tolerance test performed in adult zebrafish. Blood glucose was measured over time in control (non-ablated, DMSO) and NFP-treated (ablated) fish after intraperitoneal injection of 0.5 mg/µl of D-Glucose. 4≤ N ≤ 9 per time point for CTL and NFP. Two-way ANOVA test with Sidak's multiple comparison test. Mean ± SEM; *ns*: not significant.

The online version of this article includes the following source data for figure 4:

**Source data 1.** Cell quantification in the pancreatic tail.

**Source data 2.** Blood glucose values (glucose tolerance test).

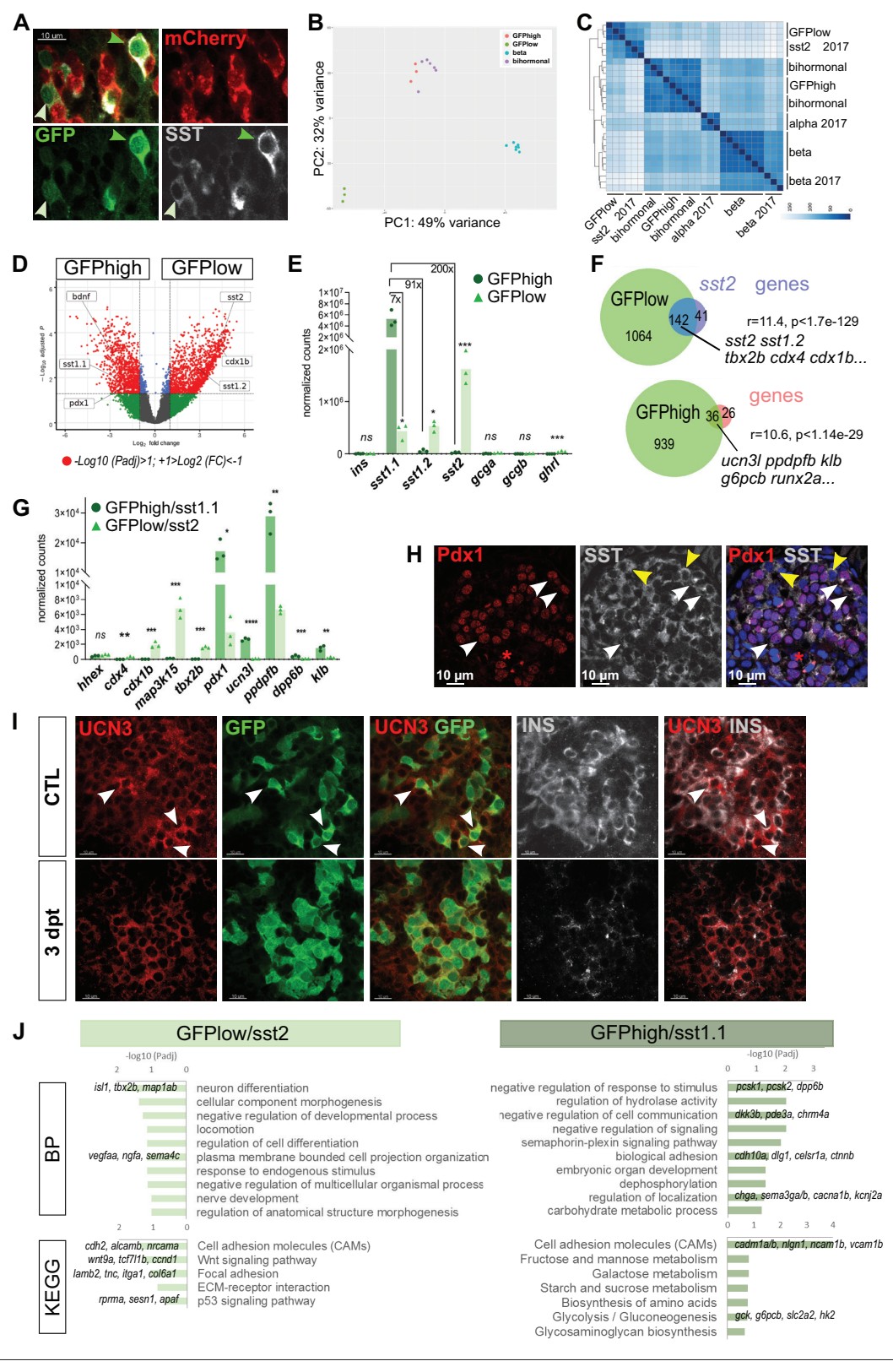

**Figure 5.** sst1.1 δ-cells (GFP^high) constitute a δ-cell subpopulation distinct from sst2 δ-cells (GFP^low) that presents similarities with β-cells. (**A**) Whole mount immunodetection on t *Tg(sst1.1:eGFP); Tg(ins:NTR-P2A-mCherry)* main islets of GFP (green), mCherry (red) and Sst (gray) revealing two levels of GFP expression (green light and dark arrowheads) that parallel the expression level of Sst. These cells are mCherry negative. (**B**) PCA plot showing the

*Figure 5 continued on next page*

*Figure 5 continued*

separation between *sst1.1*:GFP^high (n = 3), sst1.1:GFP^low (n = 3), bihormonal (n = 6) and β-cells (n = 7) based on their transcriptomic profile. 49% of the variance is explained in PC1. PCA analysis failed to separate bihormonal and *sst1.1*:GFP^high cells while separated well β-cells from the sst1.1:GFP^low cells. The *sst1.1*:GFP^high/bihormonal cluster located between β-cells and sst1.1:GFP^low cells shows that β-cells are more similar to *sst1.1*:GFP^high/bihormonal cells. (**C**) Heatmap plot showing the clustering of the sst1.1:GFP^high and sst1.1:GFP^low populations, the bihormonal cells, the β-cells of the present study and the previously published data for β-, α-, and *sst2* δ-cells (n = 3) (***Tarifeño-Saldivia et al., 2017***). In addition to revealing the expected clustering between both RNAseq data from β-cells (***Tarifeño-Saldivia et al., 2017***) and this study, this plot also shows the clustering of the GFP^low cells together with *sst2* δ-cells. (**D**) Volcano plot showing the distribution of genes expressed in GFP^high and GFP^low populations. The x-axis represents the $\log_2$ of fold change (FC) and the y-axis the $\log_{10}$ of adjusted p value (Padj) provided by DESeq. The list of all DE genes is provided in ***Figure 5—source data 1***. (**E**) Expression of the main pancreatic hormones in GFP^high and GFP^low populations (mean normalized reads). Expression is expressed as normalized counts and Padj are calculated by DESeq. *ns:* no significant DE between the two conditions, 0.05<* < 0.005, 0.0005<*** < 0.00005. (**F**) Venn diagram showing the overlap between genes overexpressed in GFP^low cells (versus GFP^high) and *sst2* δ-cell markers previously identified, and between genes overexpressed in GFP^high cells (versus GFP^low cells) and β-cell genes (***Figure 5—source data 2***). Representation factor and p value calculated by Fisher's exact test. (**G**) Expression of selected β- and *sst2* δ-cell genes in each replicate of GFP^high and GFP^low cells. GFP^high cells distinctly express high levels of *sst1.1* and will be referred to as GFP^high/*sst1.1* δ-cells, and GFP^low to GFP^low/*sst2* δ-cells. 0.05<* < 0.005, 0.005<** < 0.0005, 0.0005<*** < 0.00005, **** < 0.00001 (**H**) Confocal images showing immunodetection of Pdx1 (anti-Pdx1, red) and Sst (anti-SST, gray) on paraffin section through the main islet of a non-ablated adult fish showing double Pdx1+ Sst + cells (white arrowheads) and Pdx1- Sst+ cells (yellow arrowheads). Red asterisks highlight Pdx1 single positive cells β-cells. (**I**) Confocal images showing whole mount immunodetection of Ucn3 (red), GFP (green) and Ins (gray) in CTL and three dpt main islets from *Tg(sst1.1:eGFP); Tg(ins:NTR-P2A-mCherry)* adult fish. In CTL islets, strong Ucn3 labeling is detected in β-cells as well as in some *sst1.1*:GFP cells (white arrowheads). After β-cell ablation, Ucn3 is principally expressed in GFP+ cells that also harbor faint Ins staining. (**J**) Biological Processes (BP) and KEGG pathways overrepresented in GFP^high/*sst1.1* δ-cells (UP) compared to GFP^low cells (DOWN) (Padj<0.25). Gene Ontology (GO) terms were identified by WebGestalt (***Liao et al., 2019***) using the list of DE genes between GFP^high/*sst1.1* δ-cells and GFP^low/*sst2* δ-cells obtained with at least twofold differential expression and Padj <0.05 provided by DESeq. The list of all BP and KEGG pathways below FDR 0.25 is given in ***Figure 5—source data 4***, ***Figure 5—source data 5***.

The online version of this article includes the following source data and figure supplement(s) for figure 5:

**Source data 1.** Differential gene expression between sst1.1:GFPhigh and GFPlow cells.

**Source data 2.** List of the beta cell markers expressed in sst1.1GFPhigh cells and of the sst2 delta cell markers expressed in GFPlow cells.

**Source data 3.** List of the sst1.1:GFPhigh markers defined in this study and updated beta cell markers.

**Source data 4.** Gene Ontology analysis of genes overexpressed in sst1.1:GFPlow cells.

**Source data 5.** Gene Ontology analysis of genes overexpressed in sst1.1:GFPhigh (sst1.1 delta) cells.

**Figure supplement 1.** *sst1.1*:GFP expression delineates two distinct δ-cell subpopulations.

(***Figure 5E***). On the opposite, *sst2* was predominant in GFP^low cells though these cells also expressed *sst1.2* and *sst1.1* at lower levels. In addition, while both populations expressed the universal δ-cell marker *hhex*, other previously identified markers of zebrafish *sst2* δ-cells such as *cdx4, tbx2b*, and *map3k15* (***Tarifeño-Saldivia et al., 2017***) were specific to GFP^low cells (***Figure 5F–G***). Indeed, more than 75% of the *sst2* δ-cell genes (enriched >4 fold based on previous data ***Tarifeño-Saldivia et al., 2017***) were also enriched in GFP^low cells (***Figure 5F*** and ***Figure 5—source data 2***). Ectopic activity of the *sst1.1:GFP* transgene in the *sst2* δ-cells was confirmed by ISH showing *sst2* probe signal exclusively in the weakest GFP+ cells (***Figure 5—figure supplement 1C***). These data show that the GFP^low population contains *sst2* δ-cells, while the GFP^high population consists of a pure and distinct δ-cell population characterized by strong *sst1.1* expression. These δ-cells will be named *sst1.1* δ-cells hereafter.

Focusing on the *sst1.1* δ-cells, we noticed high expression of *pdx1* (***Figure 5G***). In addition to being expressed in all β-cells, *Pdx1* in mammals is also expressed in a subset of δ-cells (***Piran et al., 2014***; ***Segerstolpe et al., 2016***). In zebrafish, *pdx1* is expressed in β-cells but not in *sst2* δ-cells (***Tarifeño-Saldivia et al., 2017***). In agreement with the transcriptome of *sst1.1* δ-cells, Pdx1 immunolabeling was confirmed in a subset of Sst + cells on paraffin section through the adult main islet (***Figure 5H***). Next, we investigated the expression of the 62 zebrafish 'β-cell genes'. Strikingly, most of them (36/62), such

as *ucn3l*, were found enriched in *sst1.1* δ-cells (*Figure 5F–G* and *Figure 5—source data 2*) while none was preferentially expressed in the GFP^low cells. By immunofluorescence, Ucn3 decorated β-cells in control islets and, additionally, an even more intense staining was detected in a subset of GFP^high cells. After ablation, the anti-Ucn3 also marked bihormonal cells, confirming our RNAseq data (*Figure 5I*). Based on these new transcriptomic datasets, we defined the genes selectively enriched ( > 4 fold) in *sst1.1* δ-cells versus the other endocrine cell types already available (*sst2*δ, β and α) and identified 152 specific *sst1.1* δ-cell markers, among which *bdnf*, *cdh10a*, *sox11b*, and *dkk3b* (*Figure 5—source data 3*). An updated list of 60 markers enriched in β-cells versus *sst1.1* δ-cells, α and *sst2* δ-cells altogether could also be defined. Our RNAseq data also revealed that *dkk3b* and *ucn3l*, previously attributed to β-cells, were even more enriched in *sst1.1* δ-cells.

Top GO terms overrepresented in GFP^low/*sst2* δ-cells (*Figure 5J* and *Figure 5—source data 4*) were related to neuron differentiation, adhesion and Wnt signaling. Top most significant GO terms and pathways in *sst1.1* δ-cells (*Figure 5J* and *Figure 5—source data 5*) included 'biological adhesion' and proprotein convertases important in the secretory pathway such as *pcsk1* and *pcsk2*. Together with *gck*, *g6pcb*, *slc2a2*, and *hk2* associated with 'metabolism of carbohydrates', these signatures suggest some competence of *sst1.1* δ-cells for glucose-responsiveness and hormone secretion.

Overall, these data unveil that *sst1.1* δ-cells represent a distinct δ-cell population possessing basic features of β-cells and sensors to integrate Ins signaling, glucose metabolism and carry hormone secretory activity.

## Monohormonal *sst1.1*-expressing cells transcriptionally activate the *ins* gene following β-cell ablation

The transcriptomic profile of *sst1.1* δ-cells suggests that they represent a promising candidate as cellular origin of bihormonal cells. In line with a conversion of *sst1.1* δ-cells to bihormonal cells, the number of monohormonal GFP^high cells was reduced after ablation in adult fish compared to CTL (from 979 cells to 315 at 20 dpt) (*Figure 6A*, *Figure 6—source data 1*). To test the hypothesis of a direct conversion of *sst1.1* δ-cells, we followed the appearance of bihormonal cells by in vivo time lapse imaging of the main islet in *Tg(sst1.1:eGFP); Tg(ins:NTR-P2A-mCherry)* larvae after ablation from 3 to 4 dpf. *Figure 6B–B'* show mCherry fluorescence progressively appearing in monohormonal *sst1.1*:GFP+ cells presenting strong GFP fluorescence, most likely *sst1.1* δ-cells. These results indicate the activation of the *ins* promoter of the *ins:mCherry* transgene in *sst1.1*:eGFP cells and suggest that at least some *sst1.1* δ-cells directly convert into bihormonal cells immediately after ablation.

## Bihormonal cells have a transcriptomic profile very similar to *sst1.1* δ–cells but with distinct cell cycle signatures

As the PCA and clustering analyses shown *Figure 5B–C* revealed that bihormonal and monohormonal GFP^high/*sst1.1* δ-cells are transcriptionally similar, we next directly performed a pairwise comparison of their transcriptome. This analysis revealed a few DE genes, with 293 over- and 180 underexpressed genes in bihormonal cells versus *sst1.1* δ-cells (FC twofold, Padj <0.05) (*Figure 6C* and *Figure 6—source data 2*), indicating that the identity of bihormonal cells is very close to *sst1.1* δ-cells. The *ins* gene was the top overexpressed gene in bihormonal cells (54-fold) (*Figure 6D*). Among the 293 over-expressed genes in bihormonal cells, 9 were β-cell markers such as *ins* and *fstl1a* and, among the 180 underexpressed genes, 8 were *sst1.1* δ-cell markers. Both *sst1.1* and *hhex* were equally expressed, further underscoring that bihormonal cells and *sst1.1* δ-cells have a close identity.

GO analyses of the genes overexpressed in bihormonal cells identified 'ribosome', 'proteasome', 'p53 signaling pathway', and 'cell cycle' pathways as top enriched pathways (*Figure 6E* and *Figure 6—source data 3* and *Figure 6—source data 4*). To corroborate the cell cycle signature, we examined Proliferating Cell Nuclear Antigen (PCNA) in the main islet of *Tg(ins:NTR-P2A-mCherry)* adult fish. In CTL islets, PCNA immunodetection was almost absent. In contrast, it was widely observed in mCherry+ cells at 20 dpt (*Figure 6F*). As mCherry+ cells are also bihormonal, it can be concluded that PCNA is expressed in bihormonal cells. We also examined Pdx1 as a proxy for β, *sst1.1* δ and bihor-monal cells (*Figure 6—figure supplement 1A*). The proportion of PCNA+ Pdx1+ cell was strongly increased in 3 and 20 dpt islets compared to CTL. To assess more specifically DNA replication, we performed a 2-day incorporation of the established marker of DNA synthesis EdU in *Tg(sst1.1:eGFP); Tg(ins:NTR-P2A-mCherry)* larvae (*Figure 6—figure supplement 1B*). Larval *sst1.1*:GFP+ cells and

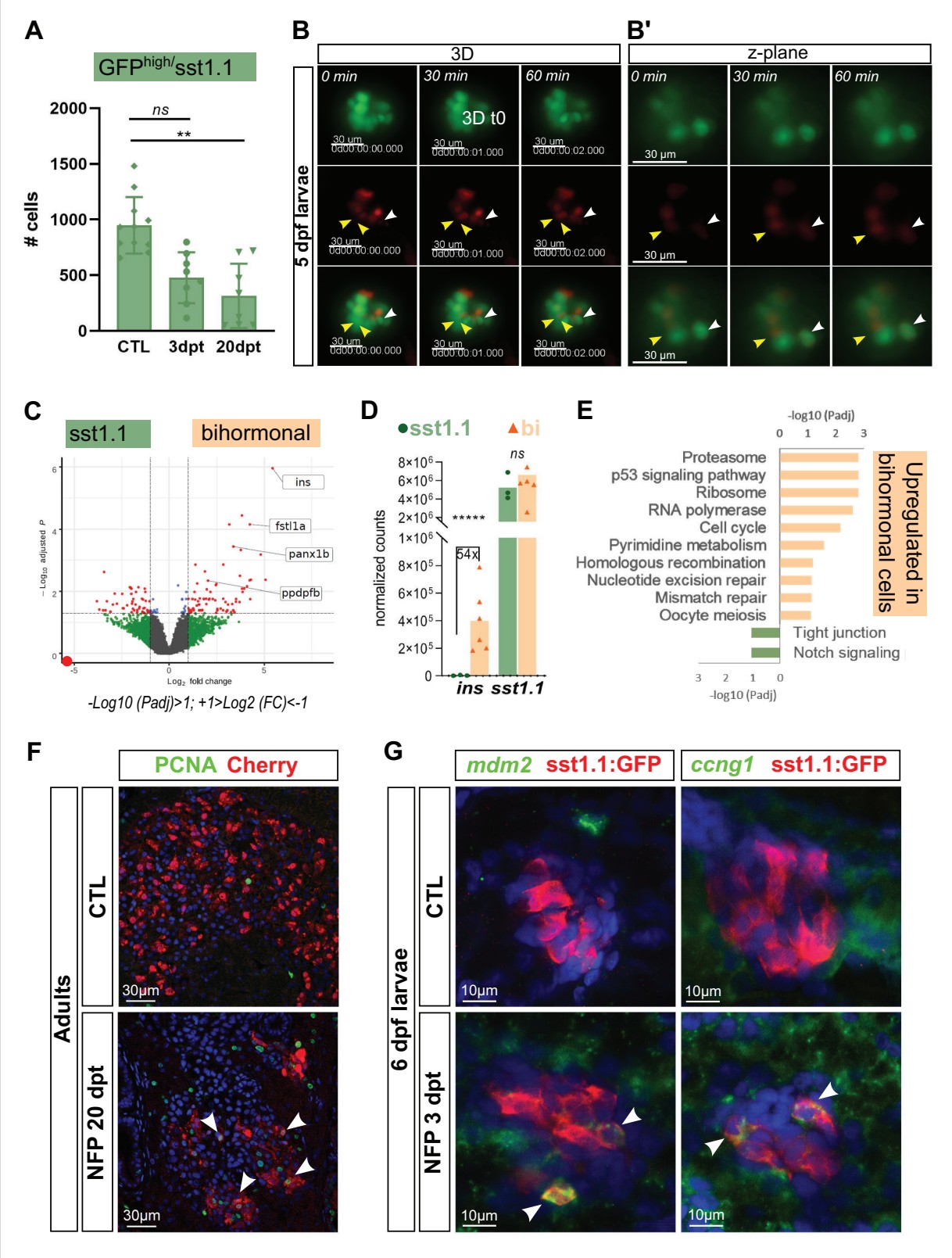

**Figure 6.** *sst1.1* δ-cells convert to Sst1.1+ Ins + bihormonal cells after β-cell destruction and activate cell cycle genes and p53. (**A**) Quantification by flow cytometry of GFP^high/*sst1.1* δ-cells before ablation (CTL) and at 3 and 20 dpt showing depletion of *sst1.1* δ-cells during regeneration. Cells were isolated from dissected main islets of adult *Tg(sst1.1:eGFP); Tg(ins:NTR-P2A-mCherry)*. Mean ± SD; Kruskal-Wallis test; *ns*: not significant, **p < *0.005* (see also **Figure 6—source data 1**). (**B**) In vivo time lapse of the main islet of a four dpf *Tg(sst1.1:eGFP); Tg(ins:NTR-P2A-mCherry)* larva following β-cell

*Figure 6 continued*

ablation from 3 to 4 dpf. 3D representation (**B**) and one z-plane (**B′**) of the same islet are shown. The arrowheads point at two GFP+ cells (green) that start to express *ins*:mCherry (red) fluorescence between *t0* and *t1* (visible in the same z-plane). The white arrowhead points to a strongly fluorescent sst1.1:GFP^high cell. Images were acquired every 30 min starting from four dpf (96 hpf). (**C**) Volcano plot showing the significant DE genes over- or underexpressed in 20 dpt bihormonal cells versus CTL GFP^high/*sst1.1* δ-cells (FC >2 < , Padj <0.05). The full list of significant DE genes calculated by DESeq is provided in *Figure 6—source data 2*. (**D**) Expression in normalized counts of the *sst1.1* and *ins* genes in CTL GFP^high/*sst1.1* δ-cells and bihormonal cells (bi). Padj are calculated by DESeq. *ns*: no significant DE between the two conditions, ***** < 0.000005. (**E**) Top significant KEGG pathways identified among the genes upregulated (in orange) and downregulated (in green) in bihormonal cells compared to CTL GFP^high/*sst1.1* δ-cells. The list of GO terms below FDR 0.25 is given in *Figure 6—source data 3*, *Figure 6—source data 4*. (**F**) Immunofluorescence of PCNA and mCherry on paraffin sections through the main islet of *Tg(ins:NTR-P2A-mCherry)* adult zebrafish, CTL and regenerated (20 dpt after NFP-mediated ablation), showing PCNA+ nuclei in mCherry+ cells in regenerated islets (confocal images, white arrowheads). (**G**) Expression of p53 target genes *mdm2* and *ccng1* mRNA (green) revealed by whole mount in situ hybridization on 6 dpf CTL and ablated *Tg(ins:NTR-P2A-mCherry); Tg(sst1.1:GFP)* larvae (main islet). Ablation was performed at 3 dpf. Immunodetection of GFP (in red) was revealed following in situ hybridization. White arrowheads point to sst1.1:GFP+ cells expressing *mdm2* and *ccng1* after ablation.

The online version of this article includes the following source data and figure supplement(s) for figure 6:

**Source data 1.** sst1.1 delta cell (GFPhigh) quantification.

**Source data 2.** Differentially expressed genes between sst1.1 delta and bihormonal cells.

**Source data 3.** Gene Ontology analysis of genes overexpessed in bihormonal cells.

**Source data 4.** Gene Ontology analysis of genes overexpressed in sst1.1 delta cells.

**Figure supplement 1.** Analysis of proliferation in the main islet of adults and larvae during regeneration.

**Figure supplement 2.** Effect on bihormonal cells of different candidate signals linked to the destruction of β-cells.

*ins*:mCherry+ β-cells displayed basal DNA replication (CTL). In NFP-treated larvae, the few mono-hormonal β-cells detected 3 days post-ablation rarely incorporated EdU showing that most escaping β-cells do not proliferate after ablation. In contrast, monohormonal GFP+ EdU + cells were observed in similar proportion between control and ablated larvae. Importantly, a significant fraction of bihormonal cells induced by the ablation showed DNA replication (*Figure 6—figure supplement 1B*).

To assess p53 activity, important for cell cycle checkpoints, we also used larvae to analyze the expression of p53 target genes by in situ hybridization. *mdm2* and *ccng1* were found induced in a subset of *sst1.1*:GFP+ cells at 3 dpt (*Figure 6G*), confirming the activation of the p53 pathway in response to the destruction of β-cells.

Given the activation of the p53 pathway following β-cell ablation, and as p53 is generally activated in response to cellular stress, we investigated the role of common stresses caused by β-cell death like hyperglycemia, oxidative stress and impaired Insulin signaling, in bihormonal cell formation. In particular, we asked whether these signals could induce by themselves the formation of bihormonal cells. However, none of these stresses was sufficient to trigger the formation of bihormonal cells (*Figure 6—figure supplement 2*).

Together, these results demonstrate that bihormonal cells in regenerating islets express genes involved in cell cycle progression and checkpoints. In line with these findings, our data also show that bihormonal cells and possibly *sst1.1* δ-cells engage in proliferation in response to the ablation of β-cells.

## Bihormonal cells also arise from pancreatic ducts

In zebrafish, the secondary islets originate from pancreatic duct-associated progenitors in a process initiated during normal larval development (*Parsons et al., 2009*; *Wang et al., 2011*). Ducts also contribute to β-cell regeneration in the adult zebrafish, providing new β-cells to the main and secondary islets (*Delaspre et al., 2015*; *Ghaye et al., 2015*). The striking observation that the vast majority of new *ins*-expressing cells are bihormonal in the entire pancreas raises the hypothesis that duct-derived Ins + cells also express Sst1.1. To explore this possibility, we used larvae, a well-established model to study β-cell regeneration from the ducts (*Ninov et al., 2013*). In this model, destruction of β-cells not only induces their regeneration in the main islet but also activates duct-associated progenitors to produce more β-cells. We first determined the time course of duct-derived β and *sst1.1* δ-cell formation during normal development and established that they start to differentiate between 7 and 10 dpf (*Figure 7—figure supplement 1*). Next, we used the *Tg(nkx6.1:eGFP); Tg(ins:NTR-P2A-mCherry)* line, where *nkx6.1* is a marker of pancreatic ducts and of duct-associated progenitors (*Ghaye et al.,*

*2015*), to perform the ablation of β-cells at 3 dpf, that is before the normal differentiation of β and *sst1.1* δ-cells in the tail. Thus, potential Ins + Sst1.1+ bihormonal cells appearing in the tail after ablation are expected to originate from the ducts and not from secondary β or *sst1.1* δ-cells. At 17 dpf, mCherry and Sst immunodetection was analyzed (*Figure 7A–B*). Double positive bihormonal cells were found in the ductal nkx6.1:GFP+ domain in the tail of regenerating larvae while they were almost absent in CTL ducts (*Figure 7B–B'–C* and *Figure 7—source data 1*). These findings support that duct cells give rise to bihormonal cells during regeneration and that they contribute to the overall bihormonal cell mass.

## Bihormonal cells persist long after β-cell ablation

Finally, we questioned the persistence of bihormonal cells long after ablation and analyzed the main islet from *Tg(sst1.1:eGFP); Tg(ins:NTR-P2A-mCherry)* adult fish 4 months after ablation. Surprisingly, most Ins + cells still coexpressed GFP as well as high levels of Ucn3 at this stage (*Figure 8A*), similarly to 20 dpt bihormonal cells. Bihormonal cells still constituted the vast majority of *ins*-expressing cells in the main islet compared to monohormonal β-cells (*Figure 8B–D*). This also suggests that they do not represent a transient intermediary population that would ultimately resolve into *ins*-only β-cells.

## Discussion

Pancreatic endocrine cell plasticity and impaired identity has emerged as an important cellular adaptive behavior in response to β-cell stress and death in human and in mammalian diabetic models. Here, we show that, in zebrafish, a large and predominant population of Ins + Sst1.1+ bihormonal cells arise after β-cell destruction, confers glucose responsiveness and restores blood glucose homeostasis. Moreover, contrasting with the age-dependent and limited β-cell neogenesis of mouse models (*Chera et al., 2014*; *Perez-Frances et al., 2021*; *Thorel et al., 2010*), bihormonal cell formation in zebrafish is fast and efficient and occurs all along life.

Our study provides an in-depth characterization of the zebrafish *sst1.1* δ-cell subpopulation. The existence of two distinct δ-cell subpopulations corroborates a recent report of two clusters of δ-cells detected by single cell RNAseq, one expressing *sst2/sst1.2* and the other *sst1.1* (*Spanjaard et al., 2018*). Although our β-cell lineage tracing experiment in larvae indicates that a subset of bihormonal cells derive from pre-existing β-cells, the majority have a non-β origin. Here, we present evidences that bihormonal cells originate from *sst1.1* δ-cells and duct cells. In contrast to *sst2* δ-cells which have previously been excluded as a source of new Ins-expressing cells (*Ye et al., 2015*), our results strongly suggest that *sst1.1* δ-cells rapidly adapt to the loss of β-cells and activate *ins* expression. First, pre-existing *sst1.1* δ-cells already express many genes essential for β-cells such as *pdx1, ucn3l* and the glucose transporter *slc2a2* (Glut2). Second, *sst1.1* δ-cells and bihormonal cells have a very close transcriptomic profile meaning that only minor changes in *sst1.1* δ-cells would generate bihormonal cells. Third, *sst1.1* δ-cells express the basic molecular machinery for glucose-sensing, glucose- and calcium-dependent stimulation of Insulin secretion and blood glucose control. Fourth, the appearance of bihormonal cells during regeneration concurs with a reduction of the *sst1.1* δ-cell mass. Finally, in vivo imaging revealed the activation of *ins* expression in *sst1.1* δcells early after ablation. All these observations support the conclusion that *sst1.1* δ-cells constitute a distinct zebrafish δ-cell population expressing β-cell features enabling them to rapidly reprogram to bihormonal cells by activating *ins* expression and engender functional surrogate β-cells. Importantly, during the preparation of our manuscript, Singh et al. also identified *sst1.1+ ins + δ/β* hybrid cells in zebrafish by scRNAseq (*Singh et al., 2022*). They also proposed the *sst1.1* δ-cells as possible cellular origin after β-cell ablation, thereby consolidating our findings. A difference between our two studies, however, is that they detected some hybrid Sst1.1+ Ins + cells in control islets while we could not clearly identify them, probably due to different technical approaches.

The fact that the bihormonal cell population is somewhat larger than the *sst1.1* δ-cell population (compare 979 GFP^high/*sst1.1* δ-cells in CTL fish in *Figure 6A* with ~1400 bihormonal cells post-ablation in *Figure 1F*) suggests the implication of mechanisms complementary to direct conversion. Indeed, beside *sst1.1* δ-cells as cellular origin of bihormonal cells, our findings also point to alternative sources, (i) a β-cell origin from pre-existing cells spared by the ablation and (ii) a ductal origin, at least in larvae. Our results show that a small but significant fraction of bihormonal cells arises from β-cells.

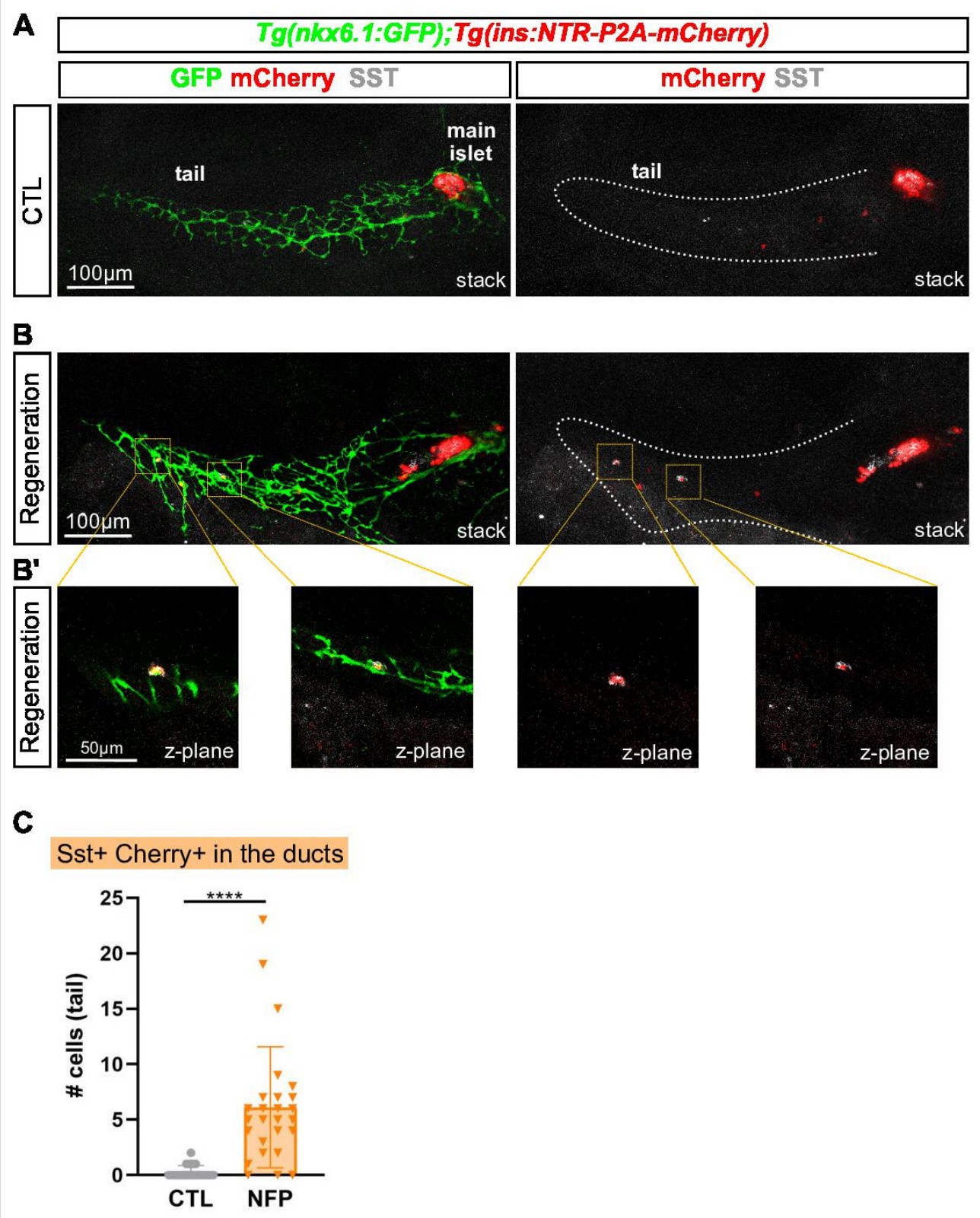

**Figure 7.** Bihormonal cells can also arise in the pancreatic ducts. (**A–B**) Whole mount immunodetection of GFP that highlights the ducts (green), mCherry (red) for β-cells and Sst (gray) on the entire pancreas of *Tg(nkx6.1:eGFP); Tg(ins:NTR-P2A-mCherry)* larvae at 17 dpf. (**A**) CTL larvae showing the main islet in the head and a few monohormonal endocrine cells (mCherry+ or Sst+) in the ductal GFP+ domain in the tail. The pancreatic tail is delineated by white dashed lines. (**B**) After treatment with NFP from 3 to 4 dpf, regenerating larvae display scattered bihormonal cells (red and gray) in

*Figure 7 continued on next page*

*Figure 7 continued*

the tail along the ducts. Stacks represent 3D projections of confocal images of the whole pancreas. (**B'**) Close-ups of two individual bihormonal cells in the tail (z-planes showing one unique optical section). (**C**) Quantification of Sst + mCherry + bihormonal cells based on confocal images. Mann-Whitney test, *****p < 0.0001*. (See also *Figure 7—source data 1*).

The online version of this article includes the following source data and figure supplement(s) for figure 7:

**Source data 1.** Bihormonal cell quantification in the tail (larvae).

**Figure supplement 1.** Time course of normal β and sst1.1 δ-cells differentiation from intrapancreatic ducts in the tail of *Tgnkx6.1:eGFP; Tg(ins:NTR-P2A-mCherry)* control larvae.

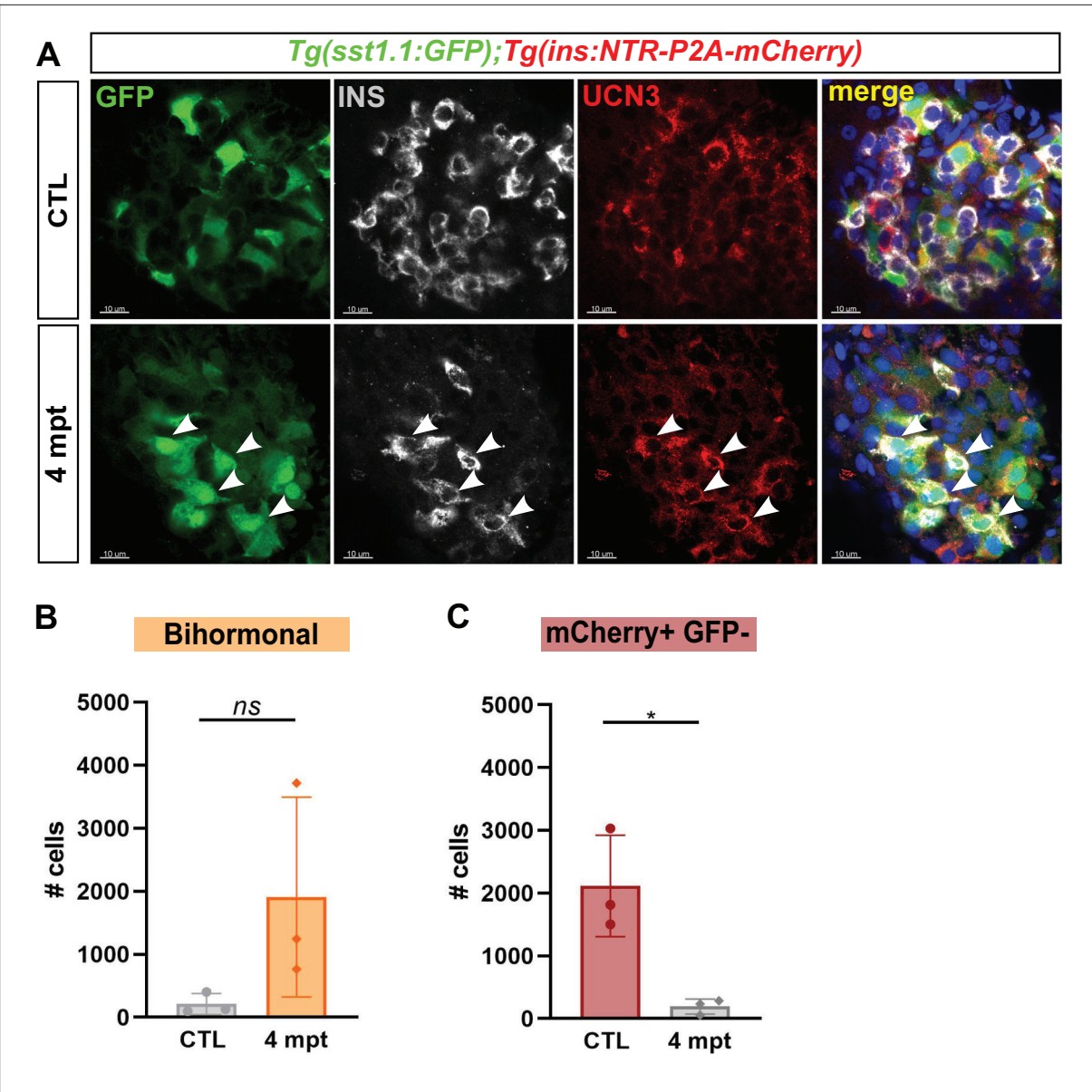

**Figure 8.** Protracted bihormonal cells 4 months after β-cell ablation. (**A**) Whole mount immunodetection of Ucn3 (red), GFP (green), Ins (grey) on the main islet of *Tg(sst1.1:eGFP); Tg(ins:NTR-P2A-mCherry)* adult fish revealing persistent bihormonal GFP+ Ins + cells still 4 months after ablation. These cells still also express Ucn3 (white arrowheads). (**B–D**) Quantification by flow cytometry of islet cell populations in CTL and 4 months after ablation. (**B**) mCherry+ GFP + bihormonal cells. (**C**) mCherry+ GFP- monohormonal β-cells. Means ± SD; Unpaired t-test with Welch's correction; *p < 0.05*.

We also show that bihormonal cells form in the pancreatic ducts. As ducts are present in the tail as well as in the head, these results suggest a ductal contribution to the global bihormonal cell mass, that is the main and secondary islets. Whether regenerating duct-derived bihormonal cells differentiate via a monohormonal *sst1.1* δ-cell transitional state remains to be determined. Moreover, the ducts could help repopulate the *sst1.1* δ-cells after conversion. Besides neogenesis, our results suggest that proliferation contributes to the formation and/or maintenance of the pool of bihormonal cells and *sst1.1* δ-cells. Notably, we observed evidences of proliferation at an early stage after β-cell ablation, 3 dpt, as illustrated by replicating EdU + bihormonal cells in larvae and broad PCNA expression in adults. Interestingly, the activation of p53 indicates a tight control on proliferation in bihormonal cells. At 20 dpt, the p53 pathway represents the second most enriched signature in bihormonal cells, while PCNA is still widely expressed. To understand this observation, it would be interesting to tackle the dynamics of cell cycle and to perform a detailed analysis of different markers of cell cycle progression and checkpoints in the different cell populations during regeneration.

The identification of bihormonal cells in zebrafish brings the question of the molecular mechanisms underlying this β/δ hybrid identity. In mammals, *Pdx1* is essential for β-cell function notably through activation of *Ins* and of the glucose-sensing machinery genes *Slc2a2* and *Gck* (*Ahlgren et al., 1998*; *Waeber et al., 1996*; *Watada et al., 1996*). *Pdx1* is also crucial to promote and maintain β-cell identity as it activates β-cell genes and represses the α-cell program (*Ahlgren et al., 1998*; *Gao et al., 2014*). Interestingly, *Pdx1*, also known as STF1 (Somatostatin Transcription Factor 1), is expressed in a subset of mouse/human δ-cells (*Piran et al., 2014*; *Segerstolpe et al., 2016*) and stimulates *Sst* expression (*Leonard et al., 1993*). In both murine α and γ-cells, the efficiency of reprogramming to *Insulin*-expressing cells is potentiated by forced expression of Pdx1 (*Cigliola et al., 2018*; *Perez-Frances et al., 2021*). Thus, the expression of *pdx1* could underlie the intrinsic competence of *sst1.1* δ-cells (or mammalian δ-cells) to induce *ins*. However, *pdx1* expression alone is obviously not sufficient to guarantee *ins* expression, and other mechanisms consequent to β-cell loss must operate in synergy, such as metabolic changes and epigenetic regulations. In contrast to *pdx1*, *nkx6.2* and *mnx1*, two genes essential for β-cell development in zebrafish (*Binot et al., 2010*; *Dalgin et al., 2011*), are totally absent in bihormonal cells (*Figure 3* and *Figure 3—source data 1*). In mammals, the homologue of *nkx6.2* in β-cells is *Nkx6.1* (see species-specific expression in *Figure 3—source data 5*). Both *Nkx6.1* and *Mnx1* genes in mouse are important to repress non-β endocrine lineage programs (*Pan et al., 2015*; *Schaffer et al., 2013*). Together, the robust expression of *pdx1* and the lack of *mnx1* and *nkx6.2* are potential key players in the hybrid β/δ phenotype.

Normal glycemia is nearly recovered after 20 days and regenerated animals display perfectly normal glucose tolerance despite the very low abundance of genuine monohormonal β-cells. Bihormonal cells formed after β-cell destruction are abundant – nearly half the initial β-cell mass – and constitute the vast majority of *ins*-expressing cells throughout the whole pancreas and hence the main source of Ins. Their capacity to regulate blood glucose levels is corroborated by their transcriptomic profile showing the expression of the machinery required for glucose responsiveness and insulin secretion as illustrated by the glucose transporter Glut2 (*slc2a2*), the prohormone convertase *pcsk1*, the $K_{ATP}$ subunit SUR1 (*abcc8*) and several components of the secretory pathway. All these findings are further supported by the observation by Singh et al that β/δ hybrid cells gain glucose responsiveness during regeneration as assessed by in vivo Calcium imaging (*Singh et al., 2022*). Altogether, we propose that, despite the fact that bihormonal cells are not identical to β-cells, they are the functional units that control glucose homeostasis in regenerated fish, compensate for the absence of monohormonal β-cells and reverse diabetes.

# Materials and methods

## Key resources table

| Reagent type (species) or resource | Designation | Source or reference | Identifiers | Additional information |
|---|---|---|---|---|
| Genetic reagent (*Danio rerio*) | *TgBAC(nkx6. 1:eGFP)<sup>ulg004</sup>* | PMID:26329351 | ZFIN: ZDB-ALT-160205–1 | |
| Genetic reagent (*Danio rerio*) | *Tg(ins:NTR-P2A-mCherry)<sup>ulg034</sup>* | PMID:29663654 | ZFIN: ZDB-ALT-171122–9 | |

*Continued on next page*

*Continued*

| Reagent type (species) or resource | Designation | Source or reference | Identifiers | Additional information |
|---|---|---|---|---|
| Genetic reagent (*Danio rerio*) | *Tg(sst1.1:eGFP)*ᵘˡᵍ⁰⁵⁴ | This paper | | See Zebrafish husbandry and generation of the *Tg(sst1.1:eGFP)*ᵘˡᵍ⁰⁵⁴ *zebrafish line* |
| Antibody | Anti-GFP (chicken polyclonal) | Aves Labs | GFP-1020 | (1:500) |
| Antibody | Anti-Insulin (guinea pig polyclonal) | Dako | A0564 | (1:500) |
| Antibody | anti-mCherry/ dsRed (Living Colors Polyclonal) | Clontech | 632,496 | (1:500) |
| Antibody | anti-Pan-RCFP (Living Colors Polyclonal) | Clontech | 632,475 | (1:500) |
| Antibody | anti-Somatostatin (rat polyclonal) | Invitrogen | MA5-16987 | (1:300) |
| Antibody | anti-Somatostatin (rabbit polyclonal) | Dako | A0566 | (1:300) |
| Antibody | anti-Glucagon (mouse monoclonal) | Sigma | G2654 | (1:300) |
| Antibody | anti-Urocortin 3 (rabbit polyclonal) | Phoenix Pharmaceuticals | H-019–29 | (1:300) |
| Antibody | Anti-Pdx1 (guinea pig polyclonal) | From Chris Wright | | (1:200) |
| Antibody | PCNA | Sigma-Aldrich | P8825 | (1:500) |
| Antibody | Goat anti-Rat IgG (H + L) Cross-Adsorbed, Alexa Fluor 488 | Invitrogen | A11006 | (1:750) |
| Antibody | Goat anti-Chicken IgY (H + L), Alexa Fluor 488 | Invitrogen | A-11039 | (1:750) |
| Antibody | Goat anti-Chicken IgY (H + L), Alexa Fluor 568 | Invitrogen | A-11041 | (1:750) |
| Antibody | Goat anti-Mouse IgG (H + L) Cross-Adsorbed Secondary Antibody, Alexa Fluor 488 | Invitrogen | A-11001 | (1:750) |
| Recombinant DNA reagent | p3E-CREᴱᴿᵀ² | This paper | | plasmid |
| Recombinant DNA reagent | p5E-MCS | Tol2kit | 228 | plasmid |
| Recombinant DNA reagent | p3E-eGFP | Tol2kit | 366 | plasmid |
| Recombinant DNA reagent | pDestTol2p2A | Tol2kit | 394 | plasmid |
| Recombinant DNA reagent | pDONRP2R-P3 | | | plasmid |
| Sequence-based reagent | O99 | This article | PCR primer | GGGGACAGCTTT CTTGTACAAAGTGG CTGCTAACCAT GTTCATGCCTTC |
| Recombinant DNA reagent | *Tg(ubb:loxP-CFP-loxP-zsYellow)* | PMID:21623370 | ZDB-TGCONSTRCT-111115–6 | |

*Continued on next page*

*Continued*

| Reagent type (species) or resource | Designation | Source or reference | Identifiers | Additional information |
|---|---|---|---|---|
| Sequence-based reagent | O100 | This article | PCR primer | GGGGACAACTTTG TATAATAAAGTTGTC AAGCTGTGGCA GGGAAACCC |
| Sequence-based reagent | IM217 | This article | PCR primer | ttttattaaagtgtttat ttggtctcagag |
| Sequence-based reagent | IM256 | This article | PCR primer | AAGAGCACTT CAGATGTCTTCCC |
| Sequence-based reagent | O097 | This article | PCR primer | GTATCTATAGTT GAACATGA AAGCAT |
| Sequence-based reagent | O098 | This article | PCR primer | GGTCACACTG ACACAAA CAC ACA |
| Sequence-based reagent | pCR8/GW/TOPO | Invitrogen | K250020 | |
| Commercial assay or kit | Gateway LR Clonase II Enzyme mix | Invitrogen | 11791020 | |
| Commercial assay or kit | Gateway BP Clonase II Enzyme mix | Invitrogen | 11789020 | |
| Commercial assay or kit | Nextera XT DNA Library kit | Illumina | FC-131–1024 | |
| Commercial assay or kit | Click-iT EdU Cell Proliferation Kit for Imaging, Alexa Fluor 647 dye | Invitrogen | C10340 | |
| Chemical compound, drug | 4-Hydroxytamoxifen | Sigma-Aldrich | H7904 | |
| Chemical compound, drug | Nifurpirinol | Sigma-Aldrich | 32,439 | |
| Software, algorithm | Flowing Software 2 | https://bioscience.fi/services/cell-imaging/flowing-software/ | RRID:SCR_015781 | Version 2.5.1 |
| Software, algorithm | Imaris | Bitplane (http://www.bitplane.com/imaris/imaris) | RRID:SCR_007370 | Version 9.5 |
| Software, algorithm | GraphPad Prism | GraphPad Prism (https://graphpad.com) | RRID:SCR_015807 | Version 8 |
| Software, algorithm | DESeq2 | DESeq2 (https://bioconductor.org/packages/release/bioc/html/DESeq2.html) | RRID:SCR_015687 | |
| Software, algorithm | WebGestalt | WebGestalt (http://www.webgestalt.org/) | RRID:SCR_006786 | |

## Zebrafish husbandry and generation of the Tg(sst1.1:eGFP)[ulg054] zebrafish line

Zebrafish wild-type AB were used in all the experiments. *TgBAC(nkx6.1:eGFP)[ulg004]* (**Ghaye et al., 2015**) and *Tg(ins:NTR-P2A-mCherry)[ulg034]* (**Bergemann et al., 2018**) were used. Zebrafish were raised in standard conditions at 28 °C. All experiments were carried out in compliance with the European Union and Belgian law and with the approval of the ULiège Ethical Committee for experiments with laboratory animals (approval numbers 14–1662, 16–1872; 19–2083, 21–2353).

To generate the *Tg(sst1.1:eGFP)[ulg054]* zebrafish line, the *sst1.1:eGFP* transgene has been generated by cloning a 770 pb PCR fragment containing the *sst1.1* regulatory regions just upstream the ATG of the *sst1.1* ORF (ENSDARG00000040799.4) amplified with primers IM217 and IM256 into the Gateway

vector pCR8/GW/TOPO. The promoter was assembled by LR recombination with p5E-MCS and p3E-eGFP into pDestTol2p2A from the Tol2kit (*Kwan et al., 2007*). Tg(sst1.1:eGFP)*ulg054* fish have been generated using the Tol2 mediated transgenesis (*Kawakami, 2007*). Adult *Tg(sst1.1:eGFP)ulg054* fish (abbreviated *Tg(sst1.1:eGFP)*) were crossed with *Tg(ins:NTR-P2A-mCherry)ulg034* to generate a double transgenic line. The insbglob:loxP-mCherry-nls-loxP-DTA construct was created by cloning a loxP-mCherry-nls loxP cassette downstream of the *ins* promoter beta-globin intron (*Ninov et al., 2013*). Subsequently, a DTA gene was cloned downstream of the last loxP site via ligation independent cloning (InFusion, Clontech). The *Tg(ins.bglob:loxP-NLS-mCherry-loxP-DTA)bns525* line (abbreviated *Tg(ins:lox-mCherry-lox-DTA)*) was generated using the Tol2 system (*Kawakami, 2007*). The *Tg(ins:CRE-ERT2)* has been generated by LR recombination combining p5E-MCS (*Kwan et al., 2007*), pME-ins and p3E-CREERT2 vectors into pDestTol2p2A from the Tol2kit. pME-ins was obtained by cloning into the pCR8/GW/TOPO a PCR fragment of 897 pb using the primers O097 et O098 and which contains 744 bp of the insulin promoter, the exon 1 (47 bp), the intron 1 (99 bp) and the 7 bp of exon two just upstream of the ATG. p3E-CREERT2 was obtained by BP cloning into the pDONRP2R-P3 the 2200 bp PCR fragment using the primers O99 and O100 and as template the pCREERT2 kindly received from P. Chambon (*Feil et al., 1997*).

## β-Cell ablation

Nifurpirinol (NFP) (32439, Sigma-Aldrich) stock solution was dissolved at 2.5 mM in DMSO. 4-Hydroxytamoxifen (4-OHT, H7904, Sigma-Aldrich) was dissolved in DMSO as a concentrated solution of 10 mM and kept as single-use aliquots at –80 °C. β-cell ablation in *Tg(sst1.1:eGFP); Tg(ins:NTR-P2A-mCherry)* larvae was induced by treatment with 4 µM NFP in E3 egg water. Adult fish were treated in fish water with 2.5 µM NFP. Control treatments consisted of E3 containing 0.16% DMSO. Larvae and adults were treated for 18 hr in the dark.

To induce β-cell ablation with *Tg(ins:lox-mCherry-lox-DTA); Tg(ins:CRE-ERT2)* line, larvae were treated at 7 dpf with 5 µM 4-OHT at in the dark during 2 × 2 hr with replacement with fresh 4-OHT. Larvae were then washed several times with E3 egg water to eliminate 4-OHT and allowed to regenerate.

## β-Cell tracing in larvae

β-cell labeling was performed in *Tg(ins:CRE-ERT2); Tg(ubb:loxP-CFP-loxP-zsYellow); Tg(sst1.1:GFP); Tg(ins:NTR-P2A-mCherry)* larvae at 6 dpf by 2 × 2 hr 5 µM 4-OHT before several washes in E3 egg water. At 7 dpf, β-cells were ablated with NFP and larvae were allowed to regenerate until 14 dpf before fixation.

## Intraperitoneal glucose tolerance test and blood glucose measurements

Adult fish were fasted for 24 hr then euthanized with tricaine and the glycemia was immediately measured using the Accu-Chek Aviva glucometer (Roche Diagnostics) with blood collected at the tail.

D-Glucose was dissolved in PBS at 0.5 mg/µl. After anesthesia, adult fish were injected intraperitoneally at 1 mg/g fish weight with tricaïne as described in *Eames et al., 2010*.

## 5-Ethynyl-2′-deoxyuridine (EdU) incorporation assay

Zebrafish larvae were incubated in 4 mM EdU dissolved in fish E3 water for 2 days, with replacement of the solution after 24 hr, the were euthanized in tricaine and fixed in 4% PFA. EdU was detected according to the protocol of Click-iT EdU Cell Proliferation Kit for Imaging, Alexa Fluor 647 (ThermoFisher C10340) and processed for whole mount immunodetection.

## Immunodetection of paraffin Sections

Samples were fixed and processed for immunofluorescence as previously described (*Ghaye et al., 2015*).

## Whole-mount immunodetection

Larvae were euthanized in tricaine and fixed in 4% PFA at 4 °C for 24 hr before IHC. After depigmentation with 3% H2O2/1% KOH during 15 min, larvae were permeabilized 30 min in PBS/0.5% Triton X-100 and incubated for 2 hr in blocking buffer (4% goat serum/1% BSA/PBS/0.1% Triton X-100). Primary and

secondary antibodies were incubated at 4 °C overnight. Adult fish (6–10 months) were euthanized and fixed for 48 hr. Digestive tracts were dissected, dehydrated and stored in 100% methanol at –20 °C. Before IHC, the samples were permeabilized in methanol at room temperature for 30 min, placed 1 hr at –80 °C then back at room temperature. After rehydration in PBS/0.05% Triton X-100, depigmentation was performed for 15 min followed by incubation in blocking buffer containing 4% goat serum /1% BSA/PBS/0.01% Triton X-100. The primary antibodies were incubated for 48 hr on adult samples and overnight on larvae, followed by overnight incubation with the secondary antibodies overnight at 4 °C. Primary antibodies: Anti-Insulin (guinea pig, 1:500, Dako A0564), Living Colors Polyclonal anti-mCherry/dsRed (rabbit, 1:500, Clontech 632496), Living Colors Polyclonal anti-Pan-RCFP (rabbit, 1:500, Clontech 632475), anti-GFP (chicken, 1:1000, Aves lab GFP-1020), anti-Somatostatin (rat, 1:300, Invitrogen MA5-16987), anti-Somatostatin (rabbit, 1:300, Dako, A0566), anti-Glucagon (mouse, 1:300, Sigma G2654), anti-Urocortin 3 (rabbit, 1:300, Phoenix Pharmaceuticals H-019–29), anti-Pdx1 (guinea pig, 1:200, kind gift from Chris Wright, Vanderbilt University), anti-PCNA (clone PC10 Sigma P8825). Secondary antibodies: Alexa Fluor-488,–568, –633 (goat, 1:750, Molecular Probes).

## Whole-mount in situ hybridization on embryos

The *sst1.1* and *sst2* probes were described in *Devos et al., 2002*. The *ins* probe has been described in *Milewski et al., 1998*. Fluorescent in situ hybridization were performed as described in *Tarifeño-Saldivia et al., 2017* on 3 or 6 days post fertilization embryos (dpf). The antisense RNA probes were revealed using tyramide-Cy3 followed by immunodetection of GFP.

Images of immunodetection and in situ hybridization were acquired with a Leica SP5 or a Zeiss LSM880 confocal microscope, and processed with Imaris 9.5 (Bitplane) for visualization.

## In vivo imaging

In vivo imaging was performed with a Lightsheet Zeiss Z1 microscope using a 20 x water immersion objective and 488 nm and 561 nm lasers. *Tg(sst1.1:eGFP); Tg(ins:NTR-P2A-mCherry)* larvae were treated from 1 dpf with 1-phenyl 2-thiourea (0.003% (w:v)) to inhibit pigment synthesis. After ablation with NFP from 3 to 4 dpf, larvae were anesthetized, embedded in 0.25% low melting agarose containing and mounted into FEP capillaries. Images were acquired every 30 min and were maintained during the whole experiment at 28° and with 100 ml/L tricaine. Images were converted with Imaris 9.5 (Bitplane) for visualization.

## Flow cytometry and FACS

The zebrafish pancreas contains one main big islet in the head and several smaller secondary islets in the tail. The main islets from 2 to 4 pancreata of *Tg(sst1.1:eGFP); Tg(ins:NTR-P2A-mCherry)* adult fish (6–10 months old, males and females) were dissected under epifluorescence to eliminate a maximum of non-fluorescent surrounding exocrine tissue, collected and washed in HBSS without $Ca^{2+}/Mg^{2+}$. Live cell dissociation was performed in Tryple Select 1 x solution (GIBCO) supplemented with 100 U/mL collagenase IV (Life Technologies 17104–019) and 40 µg/mL proteinase K (Invitrogen, 25530031) for 10 min at 28 °C, and stopped with 15% FBS. The GFP+ cells, mCherry+ cell and double GFP+ mCherry + cells were selected according to gates as shown in *Figure 1—figure supplement 1* (dashed lines) on FACS Aria III and sorted under purity mode and after exclusion of the doublets. The purity of the sorted cells was confirmed by epifluorescence microscopy (~95 %). Cells (about 1000–5000/fish depending on the cell type) were immediately lysed with 0.5% Triton X-100 containing 2 U/µl RNAse inhibitor and stored at –80 °C. Similar strategy was followed for cell quantification in secondary islets present in the pancreatic tail. The pancreas was dissected excluding the anterior most part containing the main islet and whole posterior tissues were dissociated and analyzed.

## Cell quantification in adults by flow cytometry

The percentage of mCherry+, GFP+ and double mCherry+ GFP + fluorescent cells in the dissociated islets was inferred from flow cytometry experiments in each quadrant delimiting negative and positive fluorescence. FACS plots were generated by FlowJo 10.6.2 and quantifications were performed using Flowing Software 2.5.1.

## mRNA sequencing of FACSed cells and bioinformatic analyses

cDNAs were prepared from lysed cells according to SMART-Seq2.0 (*Picelli et al., 2014*) for low input RNA sequencing and libraries were prepared with Nextera DNA Library kit (Illumina). Independent

biological replicates of each cell type sequenced using Illumina HiSeq2500 and obtained ~20 million 75 bp single-end reads (seven replicates for β-cells, 6 for 20 dpt bihormonal cells, 3 for sst1.1GF-P[high], 3 for sst1.1GFP[low]). Reads were mapped and aligned to the zebrafish genome GRCz11 from Ensembl gene annotation version 92 using STAR (*Dobin et al., 2013*). Gene expression levels were calculated with featureCounts (http://bioinf.wehi.edu.au/featureCounts/) and differential expression determined with DESeq2 (*Love et al., 2014*). Expression values are given as normalized read counts. Poorly expressed genes with mean normalized expression counts <10 were excluded from the subsequent analyses. DESeq2 uses Wald test for significance with posterior adjustment of P values (Padj) using Benjamini and Hochberg multiple testing. The differentially expressed (DE) genes identified with a Padj cutoff of 0.05 and fold change above two were submitted for GO analysis using WebGestalt tool (*Liao et al., 2019*).

The genes enriched in β-cells and *sst2δ*-cells above fourfold were identified using sequences obtained previously (*Tarifeño-Saldivia et al., 2017*) with prior mapping on the more recent GRCz11 v92 assembly of the zebrafish genome; they thus slightly differ from the gene list previously published (provided in *Figure 3—source data 2*). Then, new enrichment was updated to take into account the new transcriptomic data obtained for *sst1.1δ*-cells from *Tg(sst1.1:eGFP)* and the new β-cells from *Tg(ins:NTR-P2A-mCherry)* (presented in *Figure 3—source data 3*).

## Statistical Analyses

Graphs and statistical analyses were performed using GraphPad Prism 8. Data are represented as Mean ± SD except in *Figure 4C* where Mean ± SEM are shown. The statistical tests are described in the legend of the Figures.

## Acknowledgements

The authors thank the GIGA technology platforms GIGA-Zebrafish, GIGA-Genomics and GIGA-Imaging. The authors also thanks Chris Wright for providing the Pdx1 antibody.

# Additional information

## Competing interests

Didier YR Stainier: Senior editor, eLife. The other authors declare that no competing interests exist.

## Funding

| Funder | Grant reference number | Author |
|---|---|---|
| Chilean National Agency for Research and Development (ANID), Becas Chile | Scholarship | Claudio Andrés Carril Pardo |
| Belgian National Fund for Scientific Research | FRIA PhD fellowship | Laura Massoz David Bergemann Arnaud Lavergne |
| National Belgian Funds for Scientific Research | EoS Program | Marie A Dupont Jordane Bourdouxhe |
| European Regional Development Fund | Biomed Hub Technology Support | Arnaud Lavergne |
| National Belgian Funds for Scientific Research | | Bernard Peers Isabelle Manfroid Marianne M Voz |
| Belgian National Fund for Scientific Research | 30826052 | Marie A Dupont Jordane Bourdouxhe |

| Funder | Grant reference number | Author |
|---|---|---|
| Chilean National Agency for Research and Development (ANID), Becas Chile | 72170660 | Claudio Andrés Carril Pardo |
| European Regional Development Fund (Biomed Hub Technology Support) | 2.2.1/996 | Arnaud Lavergne |

The funders had no role in study design, data collection and interpretation, or the decision to submit the work for publication.

## Author contributions

Claudio Andrés Carril Pardo, Conceptualization, Formal analysis, Investigation, Methodology, Software, Validation, Visualization, Writing – review and editing; Laura Massoz, Marie A Dupont, Conceptualization, Formal analysis, Investigation, Methodology, Validation, Visualization; David Bergemann, Formal analysis, Investigation, Writing – review and editing; Jordane Bourdouxhe, Conceptualization, Data curation, Formal analysis, Investigation, Methodology, Resources, Software, Validation, Visualization, Writing – review and editing; Arnaud Lavergne, Formal analysis, Resources, Software; Estefania Tarifeño-Saldivia, Formal analysis, Software, Writing – review and editing; Christian SM Helker, Methodology, Resources; Didier YR Stainier, Funding acquisition, Resources; Bernard Peers, Funding acquisition, Resources, Supervision, Writing – review and editing; Marianne M Voz, Funding acquisition, Investigation, Methodology, Resources, Writing – review and editing; Isabelle Manfroid, Conceptualization, Formal analysis, Funding acquisition, Project administration, Resources, Supervision, Validation, Visualization, Writing - original draft, Writing – review and editing

## Author ORCIDs

Didier YR Stainier (iD) http://orcid.org/0000-0002-0382-0026
Isabelle Manfroid (iD) http://orcid.org/0000-0003-3445-3764

## Ethics

All experiments were carried out in compliance with the European Union and Belgian law and with the approval of the ULiège Ethical Committee for experiments with laboratory animals (approval numbers 14-1662, 16-1872, 19-2083, 21-2353).

## Decision letter and Author response

Decision letter https://doi.org/10.7554/eLife.67576.sa1
Author response https://doi.org/10.7554/eLife.67576.sa2

# Additional files

## Supplementary files

• Transparent reporting form

## Data availability

RNA sequencing data have been deposited at NCBI GEO.

The following dataset was generated:

| Author(s) | Year | Dataset title | Dataset URL | Database and Identifier |
|---|---|---|---|---|
| Carril Pardo CA, Massoz L, Dupont MA, Bergemann D, Bourdouxhe J, Lavergne A, Tarifeño-Saldivia E, Helker CSM, Stainier DYR, Peers P, Voz ML, Manfroid I | 2022 | δ-cell conversion to insulin+ bihormonal cells in zebrafish | https://www.ncbi.nlm.nih.gov/geo/query/acc.cgi?acc=GSE167187 | NCBI Gene Expression Omnibus, GSE167187 |

The following previously published dataset was used:

| Author(s) | Year | Dataset title | Dataset URL | Database and Identifier |
|---|---|---|---|---|
| Tarifeño-Saldivia E | 2017 | RNAseq from the pancreatic acinar, alpha, beta and delta cells from zebrafish | https://www.ebi.ac.uk/ena/browser/view/PRJEB10140 | EBI, PRJEB10140 |

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
