## [Editor Report]

Recently, there has been a growing appreciation for the existence of cellular plasticity in the adult islet. This study probes this phenomenon by exploiting the zebrafish experimental model, which has a much higher adult regeneration capacity than in mammals. A particular novel finding from the study is the identification of a subpopulation of islet delta cells that are similar to beta cells at the transcriptional level and can convert into an insulin/somatostatin co-expressing cell population upon beta cell ablation. The findings will be of particular interest to researchers interested in islet cell biology and pancreatic endocrine cell reprogramming; it will be interesting to explore whether similar delta cell subpopulations exist in mammalian islets to serve as an alternative source of insulin producing cells.

---

## [Decision Letter]

**Decision letter after peer review:**

Thank you for submitting your article "A δ-cell subpopulation with pro-β cell identity confers efficient age-independent recovery in a zebrafish diabetes model" for consideration by *eLife*. Your article has been reviewed by 2 peer reviewers, one of whom is a member of our Board of Reviewing Editors, and the evaluation has been overseen by Marianne Bronner as the Senior Editor. The reviewers have opted to remain anonymous.

The reviewers have discussed their reviews with one another, and the Reviewing Editor has drafted this to help you prepare a revised submission. A major issue is the lack of genetic lineage tracing to support the major conclusion of the study. There is good use of a series of transgenic reporter lines but the main claim of the authors of a rapid conversion of delta cells into insulin/somatostatin bihormonal cells is not supported by lineage tracing experiments, and alternative explanations for the rapid appearance of these bihormonal cells are insufficiently explored. The underlying mechanisms may be unique to zebrafish.

Essential revisions:

1. To unequivocally demonstrate the GFP^high^-sst1.1 cells become the bihormonal cells, the authors will need to perform genetic lineage analysis. Alternatively, they should dampen their conclusions and discuss alternative explanations.

2. Provide functional data on the Sst1.1/insulin bihormonal cells to demonstrate that they are capable of insulin secretion, which is presumed to underlie the recovery from hyperglycemia.

3. For the lineage tracing data using the zsYellow label in conjunction with an inducible beta cell specific Cre driver strain, explain why this experiment was done in developing embryos instead of during the adult stage where they made the original observation of the appearance of bihormonal cells that is associated with normalization of glucose levels.

4. For the studies suggesting bihormonal cells do not arise from pre-existing beta cells, please indicate the stages and timing the experiment was performed.

5. Include the "data not shown"

6. The potential role of the p53 pathway is not convincing and is based on a single experiment. This data should be augmented or perhaps weaken the interpretation.

7. Pdx1 (also known as STF1 for Somatostatin Transcription Factor) is in both beta and delta cells. In addition, there is older literature suggesting that beta and delta cells are in a common lineage, separate from the other islet cell types. This information should be presented in the introduction and much more discussion about the implications of this study's findings in the discussion.

8. The writing could be simplified in the Results section as indicated by reviewer 1 and 2 so that the authors are not over-extrapolating their RNA seq data without other supporting evidence that tests for function or expression at the protein level.

9. Throughout the manuscript, the authors refer to cell specific expression of several factors – which in some cases are either incorrect or represent species differences between zebrafish and mouse/humans (ie. Nkx2.2 is not in mammalian delta cells and therefore is not pan-endocrine; Nkx6.1 is the beta cell TF in mouse/human islets and therefore is not a "progenitor marker"; Foxo1 is in beta and delta cells). It would be extraordinarily useful for the reader to include a table of these factors and indicate their normal (islet cell expression in zebrafish vs mammals).

*Reviewer #1 (Recommendations for the authors):*

1. For the studies suggesting bihormonal cells do not arise from pre-existing beta cells, please indicate the stages and timing the experiment was performed.

2. The extensive description of the gene expression changes and GO analyses (page 10) is not very useful. It would be more informative to summarize the take away message from these gene expression changes and refer to the figure.

3. To unequivocally demonstrate the GFP^high^-sst1.1 cells become the bihormonal cells, the authors will need to perform genetic lineage analysis. Alternatively, they should dampen their conclusions and discuss the caveats.

4. The potential role of the p53 pathway is not convincing. This data should be augmented or perhaps weaken the interpretation.

5. Pdx1 (also known as STF1 for Somatostatin Transcription Factor) is in both beta and delta cells. In addition, there is older literature suggesting that beta and delta cells are in a common lineage, separate from the other islet cell types. This information should be presented in the introduction and much more discussion about the implications of this study's findings in the discussion.

6. Throughout the manuscript, the authors refer to cell specific expression of several factors – which in some cases may represent species differences between zebrafish and mouse/humans (ie. Nkx2.2 is not in mammalian delta cells and therefore is not pan-endocrine; Nkx6.1 is the beta cell TF in mouse/human islets and therefore is not a "progenitor marker"; Foxo1 is in beta and delta cells). It would be extraordinarily useful for the general audience to include a table of these factors and indicate their normal (islet cell expression in zebrafish vs mammals).

*Reviewer #2 (Recommendations for the authors):*

The writing could be simplified in certain places especially for paragraphs (e.g., lines 208-222) where they just list the various genes that are differentially expressed in their bihormonal cells vs regular beta cells. It's fine to do so in the discussion to add context about how their findings fit into the big picture goal of the field, but I feel listing gene names and a brief statement of their function in beta cells do nothing for the Results section. It instead makes the authors seem like they are extrapolating too much from their RNA seq characterization without other supporting evidence that tests for function or expression at the protein level.

There is too much conjecture on the basis of gene expression profiles and these conclusions are too strongly worded. There are interesting points made about the involvement of the p53 signaling pathway, but this is based on a single experiment.

The lack of functional data on the Sst1.1/insulin bihormonal cells would have been helpful to demonstrate that they are indeed capable of insulin secretion, which is presumed to underlie the recovery from hyperglycemia.

Lineage tracing of delta cells is absolutely necessary. It is possible that the authors are not seeing rapid conversion, but persistence of dead beta cells.

The manuscript is rather broad in its introduction and at times superficial. As an example: the extensive discussion on hub cells, while an interesting cell type, is just one example of a topic that is not directly related to the topic here.

The positioning of zebrafish as ancestral to mammals is misplaced. Both are contemporary groups of vertebrates that have diverged early in the vertebrate lineage. Observing plasticity in current teleostean species should not be taken to imply that this mechanism was already present in the shared teleost-tetrapod ancestor.

The discussion on which Sst is the functional equivalent of Sst in mammals is speculative and not directly relevant here. It could be addressed by experiments outside the scope of this study.

---

## [Author Response]

Essential revisions:1. To unequivocally demonstrate the GFP^high^-sst1.1 cells become the bihormonal cells, the authors will need to perform genetic lineage analysis. Alternatively, they should dampen their conclusions and discuss alternative explanations.

To unambiguously determine if the bihormonal cells derive from sst1.1 delta cells, we generated sst1.1:CRE-ERT2 fish (using the same promoter as for our sst1.1:GFP). Unfortunately, the 2 independent lines we obtained (2 independent insertions) were both characterized by broad silencing of the CRE transgene and only a small fraction of the sst1.1 cells expressed CRE (checked by CRE mRNA expression by in situ hybridization in embryos). A fortiori, we could barely detect recombined szYellow+ (tracer marker) cells after breeding with our ubb:loxP-CFPStop-loxP-szYellow line and treatment with 4-OHT.

As we could not genetically establish the origin of bihormonal cells, we now dampen our conclusions about the sst1.1 delta-cells and discuss more deeply two alternative cellular sources, the beta cells and ductal cells. Indeed, we agree with Reviewer 2 that the beta cell origin of bihormonal cells cannot be firmly excluded on the basis of our beta cell lineage tracing (see Response to Reviewer 2). Our results rather indicate that a fraction of bihormonal cells arise from pre-existing beta cells. In addition, we provide evidence that bihormonal cells differentiate from duct-associated progenitors. Ducts are a well-established source of progenitors for beta cell regeneration in zebrafish in both larvae and adults (Ninov, 2013; Ghaye 2015, Delaspres 2015). In this revised paper, we dedicate Figure 7 and Figure 7supplement 1 to this aspect.

In addition, we explored the role of proliferation in the formation of bihormonal cells and analysed cell cycle parameters (Figure 6 and Figure 6—figure supplement 1).

These findings also validate the transcriptomic profile of bihormonal cells.

Regarding the cellular origin of bihormonal cells, we would like here to refer to a parallel study by Nikolay Ninov’s group that was submitted in a coordinated manner with ours (Singh et al., bioRxiv, https://doi.org/10.1101/2021.06.24.449704) and entitled “A single-cell atlas of de novo β-cell regeneration reveals the contribution of hybrid β/δ cells to diabetes recovery in zebrafish”. In their work, using ingle cell transcriptomic analyses, they also identified the existence of “hybrid” ins+ sst1.1+ bihormonal cells in the zebrafish pancreas. Although they did not trace the sst1.1, their data also strongly point to the sst1.1 delta cells as one cellular origin of bihormonal cells after beta cell ablation. In addition, they also do not exclude a ductal origin. We discuss their main findings with respect to ours in this revised manuscript.

2. Provide functional data on the Sst1.1/insulin bihormonal cells to demonstrate that they are capable of insulin secretion, which is presumed to underlie the recovery from hyperglycemia.

To assess the secretory capacity of bihormonal cells, one way is to monitor in vivo the intracellular Calcium response in bihormonal cells upon treatment with glucose. This approach has been followed by Singh et al. on zebrafish larvae and they could demonstrate that bihormonal cells formed after ablation in larvae gain glucose responsiveness during regeneration.

Another way to assess the secretory capacity of bihormonal cells would be to measure blood levels of Insulin, or Insulin secreted from cultured islets. However, this cannot be done in zebrafish due to the very low amount of blood, cells and Insulin even in adult fish.

Thus, to address the question of the functionality of bihormonal cells, we opted for a complementary approach, ie a glucose tolerance test in adults. We showed that bihormonal cells represent the vast majority (95-99%) of all Insulin-expressing cells throughout the pancreas (see Figure 1 and new Figure 4), thereby minimizing the possibility that a putative population of pure beta cells (SST negative) would significantly contribute to regulate glycemia. The glucose tolerance test reveals that the regulation of blood glucose in regenerated fish after glucose injection is identical to CTL fish (new Figure 4). These observations strongly support that bihormonal cells are the main sources of Insulin in regenerated fish and that they are responsible for blood glucose homeostasis in the near absence of beta cells.

3. For the lineage tracing data using the zsYellow label in conjunction with an inducible beta cell specific Cre driver strain, explain why this experiment was done in developing embryos instead of during the adult stage where they made the original observation of the appearance of bihormonal cells that is associated with normalization of glucose levels.

Beta cell tracing was performed in larvae for the reason that bihormonal cells are observed at any stage, from embryos, larvae to adults, and that CRE/lox experiments are not efficient in adults. Indeed, we had to consider the prosaic reason that inducible CRE-mediated recombination by 4-OHT is faster and more efficient at young stages (Hans et al., Plos One, 2009; Mosimann et al., Development, 2011, see also Response to Reviewer 2) than at adult stages in which acceptable efficiency is hardly reached. This is now clarified in the manuscript lines 162-164.

4. For the studies suggesting bihormonal cells do not arise from pre-existing beta cells, please indicate the stages and timing the experiment was performed.

We inserted a graphical illustration of the experimental setting in Figure 2D. See also Material and Methods.

5. Include the "data not shown".

These data are now included in the manuscript:

i) A single exploratory replicate of RNAseq transcriptome of regenerated ins:Cherry+ cells (new Figure 1—source data 3) that led to the identification of very high expression of sst1.1 in regenerated Cherry+ cells.

ii) To assess the role of candidate stress agents in bihormonal cell formation, we now include the experiments with treatment with high concentration of glucose, ROS (H2O2) and an inhibitor of Insulin/PI3K signalling, all showing they are not sufficient to induce bihormonal cells or do not affect their formation (Figure 6—figure supplement 2).

6. The potential role of the p53 pathway is not convincing and is based on a single experiment. This data should be augmented or perhaps weaken the interpretation.

To augment the data on p53, we present now validations of the activation in the islet of the p53 pathway by in situ hybridization with ccng1 and mdm2 (shown now in Figure 6G), two established p53 target genes that were identified in our transcriptomes. We also explore the cell cycle signatures.

We decided to remove the experiment with pifithrin alpha. Indeed, using different timely treatments with the p53 inhibitor pifithrin alpha, we obtained two opposite responses: one that confirms the results shown in the first version of the paper (a decrease of bihormonal cells that is moreover paralleled by an increase of sst1.1:GFP cells), the other showing an increase. We think that p53 acts at different levels, possibly in monohormonal sst1.1 delta cells and in bihormonal cells and the understanding of these observations would be the focus of another project.

7. Pdx1 (also known as STF1 for Somatostatin Transcription Factor) is in both beta and delta cells. In addition, there is older literature suggesting that beta and delta cells are in a common lineage, separate from the other islet cell types. This information should be presented in the introduction and much more discussion about the implications of this study's findings in the discussion.

The fact that Pdx1 is expressed in at least a subset of delta cells in mouse and humans is indeed an important piece of information that we now describe in the manuscript lines 284-286 in the Results section and in the Discussion (lines 453455).

Although this observation may have led to speculate on a putative beta-delta common precursor, many studies rather indicate that key factors like Pdx1 but also Mnx1, Nkx6.1, Nkx2.2, Pax4, Pax6… are important to repress non-beta program (delta, alpha, epsilon, PP) and maintain/promote beta program in mature beta cells and/or in endocrine precursors/progenitors during development.

Here, we discuss the case of Pdx1 in beta cells, where it activates/maintains beta cell genes, while repressing the alpha cell program (Ahlgren, Genes and Dev 1998) (Gao et al. Cell Metab 2014) (lines 449 and onwards). In parallel to the expression of Pdx1 in bihormonal cells, we also discuss the case of the lack of the zebrafish equivalent of Nkx6.1, nkx6.2 in bihormonal cells, and of Mnx1 and discuss the impact of on cell identity.

We think that presenting Pdx1 in the Introduction section would anticipate too much on the results, so we chose to refer to Pdx1 in the Results and Discussion sections.

8. The writing could be simplified in the Results section as indicated by reviewer 1 and 2 so that the authors are not over-extrapolating their RNA seq data without other supporting evidence that tests for function or expression at the protein level.

We simplified and shortened the descriptive parts where we cited gene lists and tended to over-interpret the transcriptomic data.

Gene lists and GO analyses are nevertheless accessible in the Figures 3, 5 and 6 and the corresponding Source data.

We also include more experimental validations of the bihormonal transcriptomic data. These include the p53 pathway and the cell cycle signatures (new Figure 6) and the functionality of the bihormonal cells (new Figure 4).

9. Throughout the manuscript, the authors refer to cell specific expression of several factors – which in some cases are either incorrect or represent species differences between zebrafish and mouse/humans (ie. Nkx2.2 is not in mammalian delta cells and therefore is not pan-endocrine; Nkx6.1 is the beta cell TF in mouse/human islets and therefore is not a "progenitor marker"; Foxo1 is in beta and delta cells). It would be extraordinarily useful for the reader to include a table of these factors and indicate their normal islet (cell expression in zebrafish vs mammals).

We are sorry for this “zebrafish deformation”. Indeed, most of these genes display species-specific differences of expression. As suggested, to facilitate the reading of the manuscript, we include a comparative table (mammals/zebrafish) with the important transcription factors mentioned in the Results and Discussion sections (Figure 3—source data 5) and better definition of them in the corresponding text.

We decided to remove the example of nkx2.2 because, though very interesting, it is illustrative of a too speculative description of the first version. Indeed, in contrast to mouse/human Nkx2.2 that is expressed in all endocrine cells except delta, in zebrafish, nkx2.2a is expressed in all adult endocrine cell types including delta sst2 and delta sst1.1. It is, however, about 2x less expressed in bihormonal and sst1.1 delta cells compared to the other cell types. The implications of this lower – not absent – expression in bihormonal cells (and sst1.1 delta) cells in terms of cell identity are impossible to deduce without experimental testing. A potential role in maintaining/establishing a monohormonal phenotype is therefore not straightforward. We also removed foxo1 as its mRNA expression is rather ubiquitous in endocrine cells and hence does not inform on protein regulation such as nuclear localization which we did not assessed. We admit that it is not, per se, an endocrine or beta cell marker.

We explain in the table (Figure 3—source data 5) and lines 203-204 that Nkx6.1

is not expressed in zebrafish beta cells but instead is replaced by Nkx6.2. Zebrafish Nkx6.1 is considered as a progenitor marker in zebrafish, similar to murine Nkx6.1 during embryonic pancreas development (Figure 3—source data 5).

Reviewer #1 (Recommendations for the authors):1. For the studies suggesting bihormonal cells do not arise from pre-existing beta cells, please indicate the stages and timing the experiment was performed.

See our response to Essential revisions, point 4, and Figure 2D.

2. The extensive description of the gene expression changes and GO analyses (page 10) is not very useful. It would be more informative to summarize the take away message from these gene expression changes and refer to the figure.

These descriptions have been shortened and summarized. See response to Essential revisions point 8.

3. To unequivocally demonstrate the GFP^high^-sst1.1 cells become the bihormonal cells, the authors will need to perform genetic lineage analysis. Alternatively, they should dampen their conclusions and discuss the caveats.

We addressed this issue in the response to Essential Revisions point 1.

4. The potential role of the p53 pathway is not convincing. This data should be augmented or perhaps weaken the interpretation.

See response to Essential revisions, point 6.

5. Pdx1 (also known as STF1 for Somatostatin Transcription Factor) is in both beta and delta cells. In addition, there is older literature suggesting that beta and delta cells are in a common lineage, separate from the other islet cell types. This information should be presented in the introduction and much more discussion about the implications of this study's findings in the discussion.

See response to Essential revisions, point 7.

6. Throughout the manuscript, the authors refer to cell specific expression of several factors – which in some cases may represent species differences between zebrafish and mouse/humans (ie. Nkx2.2 is not in mammalian delta cells and therefore is not pan-endocrine; Nkx6.1 is the beta cell TF in mouse/human islets and therefore is not a "progenitor marker"; Foxo1 is in beta and delta cells). It would be extraordinarily useful for the general audience to include a table of these factors and indicate their normal islet (cell expression in zebrafish vs mammals).

See response to Essential revisions, point 9.

Reviewer #2 (Recommendations for the authors):The writing could be simplified in certain places especially for paragraphs (e.g., lines 208-222) where they just list the various genes that are differentially expressed in their bihormonal cells vs regular beta cells. It's fine to do so in the discussion to add context about how their findings fit into the big picture goal of the field, but I feel listing gene names and a brief statement of their function in beta cells do nothing for the Results section. It instead makes the authors seem like they are extrapolating too much from their RNA seq characterization without other supporting evidence that tests for function or expression at the protein level.

See response to Essential revisions, point 9.

There is too much conjecture on the basis of gene expression profiles and these conclusions are too strongly worded. There are interesting points made about the involvement of the p53 signaling pathway, but this is based on a single experiment.

We have drastically shortened the previous extrapolations that were simply based on transcriptomic profiles. We also provide now experimental evidences of the functionality of the bihormonal cells and validate the p53 and cell cycle signatures. See also response to Essential Revisions for p53 (point 6).

The lack of functional data on the Sst1.1/insulin bihormonal cells would have been helpful to demonstrate that they are indeed capable of insulin secretion, which is presumed to underlie the recovery from hyperglycemia.

See response to Essential Revisions (point 2).

Lineage tracing of delta cells is absolutely necessary. It is possible that the authors are not seeing rapid conversion, but persistence of dead beta cells.

See our responses hereabove in Essential revisions point 1 and about the beta cell tracing.

The manuscript is rather broad in its introduction and at times superficial. As an example: the extensive discussion on hub cells, while an interesting cell type, is just one example of a topic that is not directly related to the topic here.

We hope we improved this aspect, notably by removing excessively speculative parts and details less directly related to the topic of the paper. For example, we removed the parts on hub cells, the speculation on the SST hormones and the evolution aspect (see the 2 next remarks)

The positioning of zebrafish as ancestral to mammals is misplaced. Both are contemporary groups of vertebrates that have diverged early in the vertebrate lineage. Observing plasticity in current teleostean species should not be taken to imply that this mechanism was already present in the shared teleost-tetrapod ancestor.

Reviewer 2 is right. This part was removed from the revised paper.

The discussion on which Sst is the functional equivalent of Sst in mammals is speculative and not directly relevant here. It could be addressed by experiments outside the scope of this study.

This part was removed from the revised paper.